# Rapid retreat of permafrost coastline observed with aerial drone photogrammetry

Andrew M. Cunliffe[‡12], George Tanski[‡34], Boris Radosavljevic[5], William F. Palmer[6], Torsten Sachs[5], Hugues Lantuit[4], Jeffrey T. Kerby[7], and Isla H. Myers-Smith[2]

‡ These authors contributed equally to this work.
[1] Geography, University of Exeter, Exeter, EX4 4RJ, UK
[2] School of GeoScience, University of Edinburgh, Edinburgh, UK
[3] Faculty of Sciences, Earth and Climate, Vrije Universiteit Amsterdam, Amsterdam, The Netherlands
[4] Department of Permafrost Research, Alfred Wegener Institute, Helmholtz Centre for Polar and Marine Research, Potsdam,
Germany
[5] GFZ German Research Centre for Geosciences, Helmholtz-Centre, Potsdam, Germany
[6] Landscapes, Paris, France
[7] Neukom Institute for Computational Science, Dartmouth College, NH, USA

*Correspondence to*: Andrew M. Cunliffe (a.cunliffe@exeter.ac.uk)

**Abstract.**

Permafrost landscapes are changing around the Arctic in response to climate warming, with coastal erosion being one of the most prominent and hazardous features. Using drone platforms, satellite images and historic aerial photographs, we observed the rapid retreat of a permafrost coastline on Qikiqtaruk – Herschel Island, Yukon Territory, in the Canadian Beaufort Sea.
This coastline is adjacent to a gravel spit accommodating several culturally significant sites and is the logistical base for the Qikiqtaruk – Herschel Island Territorial Park operations. In this study we sought to (i) assess short-term coastal erosion dynamics over fine temporal resolution, (ii) evaluate short-term shoreline change in the context of long-term observations, and (iii) demonstrate the potential of low-cost lightweight unmanned aerial vehicles ('drones') to inform coastline studies and management decisions. We resurveyed a 500 m permafrost coastal reach at high temporal frequency (seven surveys over 40
days in 2017). Intra-seasonal shoreline changes were related to meteorological and oceanographic variables to understand controls on intra-seasonal erosion patterns. To put our short-term observations into historical context, we combined our analysis of shoreline positions in 2016 and 2017 with historical observations from 1952, 1970, 2000, and 2011. In just the summer of 2017, we observed coastal retreat of 14.5 m, more than six times faster than the long-term average rate of $2.2 \pm 0.1$ m a$^{-1}$ (1952-2017). Coastline retreat rates exceeded $1.0 \pm 0.1$ m d$^{-1}$ over a single four-day period. Over 40 days, we
estimated removal of ca. 0.96 m$^3$ m$^{-1}$ d$^{-1}$. These findings highlight the episodic nature of shoreline change and the important role of storm events, which are poorly understood along permafrost coastlines. We found drone surveys combined with image-based modelling yield fine spatial resolution and accurately geolocated observations that are highly suitable to observe intra-seasonal erosion dynamics in rapidly changing Arctic landscapes.

## 1 Introduction

The Arctic is the most rapidly warming region on Earth (Richter-Menge et al., 2017; Serreze and Barry, 2011). Increasing temperatures result in fundamental changes to the physical and biological processes that shape these permafrost landscapes (IPCC, 2013). Permafrost in the Northern Hemisphere is substantially degrading in many high latitude locations, resulting in direct and indirect impacts on natural systems as well as human activities and infrastructure (Schuur et al., 2015; UNEP, 2012). Coastal erosion is prevalent along Arctic coastlines in Western North American Arctic and all of Siberia, and is one of the key processes degrading permafrost (Lantuit et al., 2012). Coastal erosion mobilizes large amounts of sediment, organic matter and nutrients from permafrost (Lantuit et al., 2012; Overduin et al., 2014; Retamal et al., 2008; Wegner et al., 2015), which are released into the nearshore waters and affect marine ecosystems (Bell et al., 2016; Dunton et al., 2006; Fritz et al., 2017). Several studies have reported signs of accelerating coastal erosion rates at locations around the arctic, including the western arctic (Barnhart et al., 2014; Jones et al., 2008, 2009b; Mars and Houseknecht, 2007; Radosavljevic et al., 2016) and Siberia (Günther et al., 2013a; Kritsuk et al., 2014; Novikova et al., 2018; Ogorodov et al., 2016). However, the spatio-temporal resolution of circum-arctic studies limits inferences of wide-spread changes in coastal erosion rates (Fritz et al., 2017). Erosion plays a critical role in the longer-term evolution of permafrost coastlines (Barnhart et al., 2014) and biogeochemical cycling in coastal zones (Fritz et al., 2017; Semiletov et al., 2016; Vonk et al., 2012), and the majority of permafrost erosion studies compare multi-annual coastline changes to infer annualised erosion rates over periods of several years (Irrgang et al., 2018; Overduin et al., 2014). However, such coarse observation frequencies neglect the intra-seasonal dynamics during the open water season, including episodic thaw and abrupt erosion events. Knowledge of these intra-seasonal dynamics is essential to understand the processes and drivers controlling erosion patterns over time, and for better projecting future erosion rates in light of ongoing Arctic changes (Obu et al., 2016; Vasiliev et al., 2005).

The use of remote sensing approaches to measure changes in permafrost landscapes is increasingly common (Novikova et al., 2018). However, optical image coverage in high latitude regions has historically been widely limited to relatively coarse temporal and spatial resolutions, due to frequent cloud cover and logistical challenges that limit both satellite observations and aerial surveys (Hope et al., 2004; Stow et al., 2004). Recently there has been a widespread interest in the use of lightweight drones, also known as remotely piloted aerial systems or unmanned aerial vehicles (UAVs), to enable landscape managers and researchers to self-service their data collection needs (Klemas, 2015; Westoby et al., 2012), thus democratizing data acquisition (DeBell et al., 2015). Lightweight drones combined with image-based modelling can provide highly accurate and detailed measurements of rapidly changing features. These aerial observations can be obtained at user-determined frequencies (e.g. weekly, daily, or even hourly if weather conditions permit), using relatively inexpensive tools as suitable multirotor drones are available for less than $1,000 USD. Over the last few years, drone surveys are increasingly used for monitoring coastal systems (Casella et al., 2016; Duffy et al., 2017b; Mancini et al., 2013; Turner et al., 2016). However, there have been very few

examples of their application to monitor the ongoing rapid changes along permafrost coastlines (although see Whalen, 2017; Whalen et al., 2017).

In this study, we used repeated drone surveys to investigate short-term dynamics of an eroding permafrost coastline at Qikiqtaruk – Herschel Island (Yukon Territory) in the Canadian Beaufort Sea across a 13-month period. We investigated what additional insights are available from observing shoreline positions at fine spatial and temporal resolution, whether fine resolution observations of shoreline change could be related to meteorological and oceanographic variables, and compared intra-seasonal shoreline change with historical shoreline changes over the last 65 years. For our study area, we hypothesize that the erosion of the observed permafrost coastline adjacent to the settlement on Qikiqtaruk – Herschel Island varies greatly between years and continuing erosion could threaten key infrastructure to human activities on the island.

## 2 Methods

### 2.1 Study site

Qikiqtaruk – Herschel Island is located in the western Canadian Arctic in the Beaufort Sea (69°N, 139°W, Figure 1a). The island is an ice-thrust push-moraine formed during the maximal advance of the Laurentide ice sheet (Fritz et al., 2012; Pollard, 1990), and is underlain by ice-rich continuous permafrost (Brown et al., 1997; Lantuit and Pollard, 2008; Obu et al., 2016). Low spits composed of coarse material occur on the east and west sides of the island (Couture et al., 2018). The mean annual air temperature is -11°C (1970-2000) and the mean annual precipitation is ca. 200 mm $a^{-1}$ (Burn, 2012). The average coastal erosion rate for the whole of Qikiqtaruk – Herschel Island was 0.45 m $a^{-1}$ between 1970 and 2000 (Lantuit and Pollard, 2005; Obu et al., 2015), and 0.68 m $a^{-1}$ between 2000 and 2011 (Obu et al., 2016).

Ice-breakup typically commences in late June and open water conditions persist until early October (Dunton et al., 2006; Galley et al., 2016), although for Herschel Basin and Thetis Bay land-fast sea ice can persist for longer periods. The continental shelf in this part of the Beaufort Sea is very narrow and intersected by a deeper sea canyon, the Mackenzie Trough located north of Qikiqtaruk – Herschel Island (Dunton et al., 2006) (Figure 1b). This area is microtidal, with a mean range of just 0.15 m for semidiurnal and monthly tides, but these are superimposed on a ca. 0.66 m annual tidal cycle which peaks in late July (Barnhart et al., 2014; Huggett et al., 1975). The interaction between meteorological factors including wind and wave action and coastal morphology exert more influence on water levels than tidal cycles. The study area is characterised by dominant north-westerly (NW) and prevailing easterly (E) winds. North-westerly winds drive a positive storm surge and easterly winds drive a negative surge at Qikiqtaruk – Herschel Island (Héquette et al., 1995; Héquette and Barnes, 1990), with easterly winds also facilitating the transport of relatively warmer water discharged from the Mackenzie River towards the Island (Dunton et al., 2006). The contemporary rate of relative sea-level rise along this part of the Canadian Beaufort Sea is thought to range between ca. 1.1 to 3.5 mm $a^{-1}$ (James et al., 2014; Manson et al., 2005).

This study focusses on a 500 m long coastal stretch located to the east of Kuvluraq – Simpson Point, a coarse clastic spit (Figure 1c). The study reach is along the edge of an alluvial fan, comprised of redeposited marine and glaciogenic sediments that form Qikiqtaruk – Herschel Island (Fritz et al., 2011; Rampton, 1982). The spit is attached to the alluvial fan, and is

supplied by sediment from the alluvial fan and the high bluffs to the east (Radosavljevic et al., 2016). The focal coastline is characterized by low to moderately high bluffs (ca. 1 – 5 m in elevation). Ice contents in these bluffs are high, at typically ca. 40% ice by volume (Obu et al., 2016), which is slightly lower than the typical 50-60% ice content modelled for ice-thrust moraines along this portion of the Yukon Coast (Couture and Pollard, 2017). Permafrost temperatures are approximately -8°C (Burn and Zhang, 2009), and are known to be warming in recent decades (Burn and Zhang, 2009; Myers- Smith et al., n.d.).

This study area lies entirely within the slightly larger 'Coastal Reach 3' unit considered by Radosavljevic *et al.* (2016), who reported coastal retreat rates of $1.4 \pm 0.6$ m a$^{-1}$, $1.7 \pm 0.7$ m a$^{-1}$ and $4.0 \pm 1.1$ m a$^{-1}$ for the periods 1952-1970, 1970-2000 and 2000-2011, respectively. For further details on the changing ecological and erosional context of this site, see Burn (2012), Radosavljevic *et al.* (2016) and Myers-Smith *et al.* (n.d.).

Coastal erosion at our study site threatens the human settlement and infrastructure on Qikiqtaruk – Herschel Island, located on Kuvluraq – Simpson Point (Olynyk, 2012; Radosavljevic et al., 2016). This gravel spit bounds the natural anchorage of Ilutaq – Pauline Cove, and is an important regional hub for local and indigenous travellers, park administration and rangers, tourists, and researchers in the western Canadian Arctic (e.g. Burn and Zhang, 2009; Myers- Smith et al., n.d.). The currently seasonally-inhabited settlement is part of the Qikiqtaruk – Herschel Island Territorial Park, and accommodates a number of

culturally and historically significant sites resulting in its candidature for UNESCO World Heritage status (UNESCO, 2004). The proximity to the sea and low elevation of this settlement at $\leq 1.2$ m above sea level leads to high risk of coastal hazards, particularly flooding (Myers-Smith and Lehtonen, 2016; Olynyk, 2012; Radosavljevic et al., 2016).

## 2.2 Drone and time-lapse image acquisition

In 2016, one drone survey was conducted in late July, followed by seven additional drone surveys over a 40-day period between

July 6[th] and August 15[th] 2017. Drone surveys were conducted using two platforms: (i) a lightweight flying-wing Zeta Phantom FX-61 with a PixHawk flight controller equipped with a Sony RX-100ii camera (1" CMOS sensor with 20.2 Megapixels), and (ii) a multi-rotor DJI Phantom 4 Pro (1" CMOS sensor with 20 Megapixels). Drone operations were conducted in accordance with an SFOC issued by Transport Canada (to assist others seeking such permission, our full application is available at https://arcticdrones.org/regulations/). Black and white ground control markers were deployed along the shoreline and precisely

geolocated to an absolute accuracy of approximately 0.02 m using a real time kinematic global navigation satellite system (GNSS) equipment (Leica Geosystems). We used between 3 to 132 markers in the surveys, depending on survey extents and destruction of markers by natural processes; and ideally, we recommend using n=13 ground control markers, distributed evenly across the area of interest (Carrivick et al., 2016; Cunliffe and Anderson, 2019). Image overlap, a function of front-lap and

side-lap, captured each part of the study area in at least five and usually >10 photographs; this equates to fore-/side-lap values of 56% and 69%, respectively. For 2D orthomosaics and 3D elevation models, we ideally recommend higher levels of overlap, ca. 8-10 and ca. 12-20 overlapping images respectively. Drone surveys over this study area had flight times of ca. 15-25 minutes, at altitudes ranging from 30 m to 120 m (Table 1). The geotagged RGB photographs from each aerial survey had

ground-sampling distances ranging from 10 mm to 40 mm. Although this study presents drone surveys for a limited (500 m) extent of shoreline, drone surveys could be optimised to observe larger reaches of up to ca. 1.5 to 2 km, particularly in jurisdictions such as Canada where current regulations permit UAV operations up to 926 m from the remote pilot(s). For example, using two drones we found it possible to survey over eight km$^2$ in a single day. Survey parameters including date and time of day, aircraft, altitude and number of ground control markers are given in Table 1. For further discussion of

recommended drone survey parameters for different applications, see Carrivick *et al.* (2016) and Duffy *et al.* (2017a). Drone images were processed with structure-from-motion photogrammetry using Agisoft PhotoScan (version 1.3.3) (Agisoft, 2018; Sona et al., 2014), and processing parameters are reported in Table S1. GNSS-derived geolocations for each individual image and the precisely geolocated ground control markers provided additional spatial constraint of the photogrammetric processing (Carrivick et al., 2016; Cunliffe et al., 2016; Westoby et al., 2012). The photogrammetric processing yielded georegistered

orthomosaic composite images and digital surface models. Note that the heightfield approaches to surface modelling used in this analysis are not capable of capturing topographic change related to the undercutting of bluffs. Capturing such overhanging features with photogrammetric methods can be possible, but requires optimising image acquisition and more computationally intensive post-processing. To inform qualitative interpretation of the erosion dynamics at this location, a time-lapse camera was installed at the location indicated on Figure 1, imaging the study coastline at hourly intervals for four days between

2017-07-29 and 2017-08-03.

### 2.3 Image alignment and shoreline mapping

In addition to the drone surveys, we also used four 'historic' panchromatic aerial photographs from 1952 and 1970, and satellite images from 2000 and 2011 (previously analysed by Radosavljevic et al., 2016). These four images had already been

orthorectified in PCI Orthoengine to minimise image distortion. We co-registered these four 'historic' orthorectified images to the 2017-07-06 orthomosaic image in a geographic information system (ArcGIS, version 10.5, ESRI), as we considered this orthomosaic to have the best spatial constraint and coverage of all of the available datasets. All twelve images were aligned to a common spatial framework: NAD83 UTM 7N (EPSG: 26907). Further details of all images and composite orthomosaics are summarised in Table 1. Alignment errors estimated as the root mean square error (RMSE) of the control points. While this

approach to quantifying alignment error is standard practice in shoreline change analysis (Irrgang et al., 2018; Novikova et al., 2018; Río and Gracia, 2013), we note that the RMSE of control points is not a strong metric of this uncertainty, as transformation parameters (georeferencing) and the intrinsic and extrinsic camera parameters (structure-from-motion photogrammetry) are adjusted to minimise the RMSE. Consequently, for an independent assessment of image registration

error, in future work it would be preferable to use the RMSE of *independent* check points, which were not used to constrain transformation or bundle adjustment parameters (James et al., 2017). Visual comparison of each dataset indicated excellent spatial agreement and suitability for further analysis (Jones et al., 2018). Pixel error refers to the spatial resolution of the digital satellite and orthomosaic composite images, and for aerial photographs is a metric of image quality calculated based on the scale factor of each image multiplied by the typical resolution of a 9 x 9-in aerial photogrammetric camera (after Radosavljevic et al., 2016).

Shorelines were digitised manually in ArcGIS for all twelve images, at a scale of 1:600 for the four, older, coarser spatial resolution panchromatic images and a scale of 1:80 for the eight, finer spatial resolution red-green-blue (RGB) orthomosaics. The shoreline was defined as the vegetation edge rather than the wet-dry line previously used in this region (Radosavljevic et al., 2016), because the vegetation edge was both more visually distinct and temporally consistent than the wet-dry line (Boak and Turner, 2005). Temporal consistency was essential to ensure meaningful assessment of coastal retreat over short time intervals (Río and Gracia, 2013). Mapping shoreline edges was possible with much greater fidelity on the fine spatial resolution RGB orthomosaic images compared to the coarser spatial resolution panchromatic images where low contrast was sometimes an issue (Boak and Turner, 2005; Río and Gracia, 2013). Shoreline digitising errors were estimated by the GIS operator, and ranged between 0.1 m and 4.0 m depending on image spatial resolution (Table 1).

### 2.4. Shoreline and elevation analysis

Total shoreline uncertainties were calculated as:

$$U = E_G + E_P + E_D \qquad \text{(Equation. 1)}$$

Where U is total shoreline uncertainty, $E_G$ is georeferencing error, $E_P$ is pixel error, and $E_D$ is digitising error (Table 1) (Irrgang et al., 2018; Radosavljevic et al., 2016; Río and Gracia, 2013). Additive error propagation is appropriate because these error terms are not independent.

Shoreline position statistics were calculated with the USGS Digital Shoreline Analysis System (DSAS version 4) extension for ArcGIS (Thieler et al., 2009), using shore normal transects at 5 m intervals. Shoreline retreat rates in this study are given in end point rates for comparison between surveys, and both end point rate and linear regression rate for the entire time period of the study. The linear regression rate uncertainty is the standard error of the slope parameter at the 95% confidence interval. For further discussion on erosion rate calculation, see Thieler *et al.* (2009). The accuracy of shoreline change rates was calculated as:

$$DOA = \frac{\sqrt{U_i^2 + U_{ii}^2}}{\Delta t} \qquad \text{(Equation. 2)}$$

Where DOA is the dilution of accuracy (Dolan et al., 1991; Himmelstoss et al., 2018; Irrgang et al., 2018), $U_i$ is the total shoreline uncertainty of the first point in time (from Table 1), $U_{ii}$ is the total shoreline uncertainty of the shoreline position

from the second point in time, and $\Delta t$ is the duration of the time period in years or days, as appropriate. Erosion rate errors refer to DOA values, unless otherwise stated, and Table 2 displays the DOAs for all periods. We compared differences in modelled surface elevations between all periods across a cross-sectional transect, and across the whole study area for a 35-day period from 2017-07-06 to 2017-08-11 (dates constrained by DSM quality as discussed below). During the survey 2017-07-06, sea level was generally at -2.5 m relative to the EPSG: 26907 datum. To exclude erroneous elevation observations from the sea, we digitised the water's edge at a scale of 1:80 and assigned a constant elevation of -2.5 m to this seaward area. This approach excludes potential submarine elevation change from the subsequent volume calculations. The volume of material eroded was calculated with the surface volume tool in ArcMap.

## 2.4. Meteorological and oceanographic observations

Meteorological observations were obtained from an Environment Canada weather station located on Kuvluraq – Simpson Point (station ID: 'Herschel Island - Yukon Territory', World Meteorological Organisation ID: 71501; downloaded from http://climate.weather.gc.ca/climate_data on 2018-01-03), and processed to extract mean six-hour air temperature, wind speed, and wind direction throughout the 2017 observation period. Conductivity-temperature-depth (CTD) profiles (see supporting figure S2) were collected in July and August 2017 with a CastAway CTD (SonTek, USA) from a small research vessel ca. 1 km from the study area (near to 69.552°N, 138.923°W).

## 3 Results

### 3.1. Shoreline position analysis

Over our observational period of 1952 to 2017, the coastline along the study reach retreated by an average of $143.7 \pm 28.4$ m (where $\pm$ is the standard deviation of observations from each transect). The overall retreat rate was $2.2 \pm 0.1$ m a$^{-1}$ as calculated by end-point rate, and $1.9 \pm 0.5$ m a$^{-1}$ as calculated by the linear regression rate. Average retreat rates over decadal periods ranged between $0.7 \pm 0.3$ m a$^{-1}$ to $3.0 \pm 0.5$ m a$^{-1}$. The net shoreline change and end-point rates for all periods are presented in Table 2.

Over a 40-day period in the summer of 2017, shoreline retreat was $14.5 \pm 3.2$ m, ranging between 21.8 m to 6.1 m, at an average rate of 0.36 m per day. The observed shoreline positions are depicted in Figures 2 and 3, although the high temporal frequency of observations throughout the summer of 2017 and episodic pattern of retreat meant that the shorelines were sometimes very close in space. Most of the coastline retreat occurred over two periods: (i) 27 days between July 13[th] to July 30[th] and (ii) four days between August 11[th] to August 15[th] (Figures 2, 3, and 5, Table 2). There was minimal change in coastline position during the six days between August 5[th] and August 11[th], the seven days between July 6[th] to July 13[th], and the six days between July 30[th] and August 5[th] (Figure 2, Table 2). The erosion at this coastline over four days is illustrated in a time-lapse video (Video S1). This camera was orientated facing west by southwest (Figure 1) with the alluvial fan and the eroding cliff

in the foreground and the structures of the settlement in the background. This video illustrates the fluctuations in sea level and wave conditions and shows the undercutting, block failure and denudation of detached blocks between 2017-07-29 and 2017-08-03.

## 3.2. Surface elevation change

We generated digital surface models (DSM) of the coastal topography from all eight photogrammetric surveys undertaken in 2016 and 2017, spanning a 13-month period (Figure 4). However, in several cases, the DSMs did not yield reliable data across the entire coastal reach, due to insufficient spatial constraint of the photogrammetric reconstructions. This issue was due in part to the destruction of ground control markers by faster than anticipated coastal retreat, as well as sub-optimal distribution of GCPs and insufficient image overlap from some surveys due to weather constraints. Over the 35 days between two, well-
constrained DSMs from 2017-07-06 and 2017-08-11, ca. 28 $m^3$ $m^2$ of material was removed (totalling ca. 13 800 $m^3$ across the 500 m coastline), at an average rate of ca. 0.79 $m^3$ m $d^{-1}$ (Figure 4a). These estimates do not include the 4.1 ± 1.1 retreat observed between the 11[th] and 15[th] of August. During this four-day period, a further ca. 5 300 $m^3$ of material was probably removed (ca. 2.7 $m^3$ $m^2$ $d^{-1}$), assuming an average cliff height of 2.65 m as measured across the preceding 35 days. Combined, this resulted in an estimated 19,100 $m^3$ of material removed over the 40 days, The elevation change map (Figure 4a) illustrates
the increase in bluff elevation from ca 1 m in the west to ca. 5 m east across the coastal reach. Inland of the shoreline, there were scattered small increases in surface elevation, typically on the order of 0.1-0.2 m, these might relate to the development (esp. leafing out) of tundra vegetation during the short summer growing season (Myers- Smith et al., n.d.). In the centre of the study reach, the DSMs were of sufficient quality to allow cross-sectional comparisons across a ca. 3 m high bluff shown in Figure 4b; these cross sections were sampled across the A-B transect indicated on Figure 3 and Figure 4a. The depression in
the 2017-08-15 elevation profile corresponds with the ca. 1 m gap behind a detached block, depicted in Figure 3c, illustrating the sensitivity of the surface models.

## 3.3. Meteorological, oceanographic and time-lapse video observations

Erosion rates for each observation period through the summer of 2017 were compared to meteorological and oceanographic conditions, in order to better describe the controls on episodic and rapid erosion of this coastline (Figure 5). From three to ten
days prior to the first 2017 survey (on 2017-07-06), winds were consistently strong from the east and their six-hour average speed reached up to 40 km $h^{-1}$ (Figure 5). For zero to three days prior to the same survey the dominant wind direction shifted to the northwest with strong winds up to 40 km $h^{-1}$ (Figure 5), which raised the water level and refracted waves around the bluffs at Collinson Head to the northeast of the study reach. Over the seven days between the 6[th] and 13[th] of July 2017, winds were predominantly from the southeast, with brief periods of strong winds (ca. 30 km $h^{-1}$ over six hours) from the northwest
(Figure 5). These meteorological conditions generally promoted waves from the southeast, which eroded the exposed cliff base (Figures 4 and 5). Over the 17 days between the 13[th] and 30[th] of July, winds were variable, but predominantly from the southeast, with two notable periods of very high strength winds (ca. 40 km $h^{-1}$ for 24 and 12 hours, respectively) (Figure 5),

and surface water temperatures reached nearly 10°C (see Fig. S3, CTD profile d). These conditions combined to drive rapid erosion resulting in 7.4 ± 5.6 m (SD) of shoreline retreat in just 17 days (Figures 3 and 4).

Over the six days between the 30th of July to the 5th of August, winds were variable in direction and typically weaker (Figure 5). This resulted in minimal shoreline retreat (Table 2), but did remove cliff debris from the beach facilitating further undercutting (Video S1, Figure 4). Over the six days between the 5th to the 11th of August, the wind direction was variable and wind speed was low (6-hour averages mostly below 20 km h$^{-1}$), with relatively slow coastline retreat of ca. 0.17 m d$^{-1}$. Over the four days between the 11th and 15th of August, a larger storm event developed, with wind shifting from east through north to west and wind speeds increasing to excess of 45 km h$^{-1}$ for more than six hours (Figure 5). These meteorological conditions generated large waves and causing undercutting that drove 4.1 ± 1.1 m (SD) of shoreline retreat in just four days, largely through block failure (Figures 3, 4 and S4, Table 2). Sea surface temperatures were relatively warm at 6-10°C when measured between the 21st of July and the 2nd of August (Figure 5, Figure S2). Figure S1 summarises wind vectors and velocities observed during the summer of 2017.

## 4. Discussion

### 4.1. Rapid shoreline change

Over the 65-year record from 1952 to 2017, we found substantial erosion along the 500 m study coastline of Qikiqtaruk – Herschel Island. The average rate of retreat was 2.2 m a$^{-1}$, ranging over decadal periods from 0.7 to 3.0 m a$^{-1}$ (Table 2). This long-term retreat rate is fast compared with 0.7 m a$^{-1}$ for the Yukon coast (Irrgang et al., 2018), 1.1 m a$^{-1}$ for the Canadian Beaufort Sea (Lantuit et al., 2012), and circum-arctic observations where rates are typically between 0-2 m a$^{-1}$ (Overduin et al., 2014) with a weighted mean of 0.57 m a$^{-1}$ (Lantuit et al., 2012). Our study reach lies within the slightly larger 'coastal reach 3' unit considered by Radosavljevic *et al*. (2016); consequently, differences in reach length and historic image co-registration result in some slight differences between the erosion rates reported herein and those previously reported for the historic imagery. Coastal retreat rates in the neighbouring Alaskan Beaufort Sea were typically 0.7 to 2.4 m a$^{-1}$ depending on coast type (Jorgenson and Brown, 2005), with extremes of up to 25 m a$^{-1}$ (Jones et al., 2009b). Yet, the Alaskan Beaufort Sea coastline is more similar to the western formerly non-glaciated part of the Yukon Coast, with low cliffs, overall strong erosion rates and longer sea ice cover (Irrgang et al., 2018; Jorgenson and Brown, 2005; Ping et al., 2011). This is quite different from our study coastline in the formerly glaciated part of the Yukon Coast (Rampton, 1982), which is characterised by high cliffs and high ground ice contents due to former movement and burial of glacier ice (Couture and Pollard, 2017; Fritz et al., 2011). Furthermore, the sea-ice free season in our study area is longer than further west along the Alaska Coast due to the warming influence of the Mackenzie River, but in turn is modulated by the breakup of land-fast ice, which can be persistent in Herschel Basin and Thetis Bay (Dunton et al., 2006). Erosion rates from linear regression tend to underestimate rates calculated from end point reports (Dolan et al., 1991; Radosavljevic et al., 2016), which is consistent with our findings of 1.9 m a$^{-1}$ versus

2.2 m a$^{-1}$, but both linear regression and end point rates alone do not account for uncertainty in shoreline positions (Himmelstoss et al., 2018). Changes in the rate of mean shoreline position for all time points are shown on Figure S4.

Over a 384-day period from 27$^{th}$ July 2016 to 15$^{th}$ August 2017, we observed a large retreat in the shoreline position, with an average of 17.4 m, although note that this period is 19 days longer than a year and includes a disproportionate number of days from the open water season. Most of this rapid retreat occurred in the summer of 2017, when we measured 14.5 m of coastline retreat over just 40 days. Our own qualitative observations on the ground over the summer of 2017 (Video S1) confirmed the extremely rapid shoreline changes reported above. In this time, we estimate approximately 19,000 m$^3$ of material was eroded at a rate of ca. 0.96 m$^3$ m d$^{-1}$. The coastal erosion processes we observed during 40 days of 2017 correspond with the conceptual model described by Barnhart *et al.* (2014) (Video S1 and Figure S3). The bluffs along the alluvial fan were affected by both thermo-denudation but particularly thermo-abrasion due to the combined mechanical and thermal action of sea water causing undercutting and subsequent block failure (Barnhart et al., 2014; Günther et al., 2012; Vasiliev et al., 2005). These thermal processes are likely influenced by warm surface waters delivered from the Mackenzie River Delta during easterly wind conditions (Dunton et al., 2006). Additional factors facilitating rapid erosion at this site are the high ice content (ca. 40% Obu et al., 2016) and the low relief, as less material is deposited at the base of the bluff following cliff failure, thus reducing protection of the bluff base from further wave action (Héquette and Barnes, 1990). Given the episodic nature of coastal retreat, it can be difficult to compare short-term rate changes with long-term observation periods (<2 vs. >10 years, respectively) (Dolan et al., 1991). However, to remain consistent with the long-term average rate of 2.2 m a$^{-1}$, no further erosion of this coastline would need to occur for more than seven years after the retreat observed in 2016 and 2017.

The rapid coastline retreat observed in this study reach is consistent with, but greater than, earlier analysis of neighbouring coastal reaches on Qikiqtaruk – Herschel Island between 1952 and 2011 (Radosavljevic et al., 2016) and also coastal retreat observed in other Arctic permafrost coastlines (Günther et al., 2013b; Irrgang et al., 2018; Jones et al., 2009a; Whalen et al., 2017). Coastline retreat rates almost doubled from 7.6 m a$^{-1}$ (1955-2009) to 13.8 m a$^{-1}$ (2007-2009) at Cape Halkett on the Alaskan Beaufort Sea (Jones et al., 2009a), and more than doubled from 2.2 m a$^{-1}$ (1952-2010) to 5.3 m a$^{-1}$ (2010-2012) on Bykovsky Peninsula, Siberia (Günther et al., 2013b). Increases in erosion rates greater than two-fold are more commonly observed on low elevation coasts, such as the one examined herein and in Jones *et al.* (2009a). On the Yukon Coast, average coastal retreat rates were 0.5 m a$^{-1}$ between 1950-1970 (Harper et al., 1985) and 0.7 m a$^{-1}$ between 1950-2011 (Irrgang et al., 2018), with maximum reported rates of 22 m a$^{-1}$ on Pelly Island (NWT) 130 km to the east along the Yukon-NWT Coast (Whalen et al., 2017). Robustly detecting changes in the trends of permafrost coastline erosion in this region and more widely requires further analyses of shoreline position changes at (near-)annual temporal resolution, considering a larger range of representative coastal reaches and study sites.

## 4.2. Drivers of rapid shoreline change

The rapid retreat observed in 2016 and especially 2017 was likely driven by a range of factors, including longer term conditioning factors acting over timescales of decades to years and also shorter-term factors acting over timescales of weeks to days and hours. Over the longer term, this region experiences a relative sea level rise of ca. 1.1 to 3.5 mm a$^{-1}$ (James et al., 2014; Manson et al., 2005), progressively subjecting more permafrost to thermos-abrasional processes. Seasonal sea ice break up has advanced earlier by 46 days per decade between 2002-2016 (Assmann, 2019), with ice-free seasons lengthening by nine days per decade between 1979-2013 (Stroeve et al., 2014) and summer minimum sea ice concentrations decreasing over the last 39 years in this area (Myers- Smith et al., n.d.). This region generally is experiencing longer open water seasons and increasing wave heights (Barnhart et al., 2014; Farquharson et al., 2018; Stroeve et al., 2014), and there is increased heat influx to the ocean during the open water season, due to increasing discharge from the Mackenzie River and atmospheric warming. Atmospheric warming has increased permafrost temperatures and deepened the active layer at sites just 1 km from the study reach (Burn and Zhang, 2009; Myers- Smith et al., n.d.), lowering the energy required to thaw the permafrost, although lengthening open water seasons is likely to most significant factor (Farquharson et al., 2018).

Attribution of the rapid change in shoreline position in 2017 to a single main driver is not possible with the available datasets. Examination of MODIS observations indicates sea ice break up was ca. 15 days earlier in 2015 and 2016, but in 2017 it was in line with the 10-year average (Figure S5). The direction and frequency of wind patterns observed in 2017 (Figure 5, Figure S1) are similar to those reported in June-Sept from 2009 to 2012 (Radosavljevic et al., 2016, figure 4 therein). However, overall wind speeds were higher in 2017, with a greater proportion of periods with mean speeds in excess of >30 km h$^{-1}$ (Figure S1). The role of wind is discussed further below. A large portion of beach and cliff debris appeared to have been removed between our survey in 2016 (2016-07-26) and our first survey in 2017 (2017-07-06) (Figure 4b, and corroborated by our field observations), potentially either during storm events in the autumn of 2016 and/or ice bulldozing during ice-breakup in spring 2017. We hypothesise that removal of this protective material *may* have increased the susceptibility of these cliffs to rapid erosion in the summer of 2017. Field observations from 2018 suggest that the shoreline retreat has stabilised at rates closer to the long term average.

Through the summer of 2017, coastal retreat was highly episodic. The main mode of erosion was block failure following thermo-abrasional undercutting. This undercutting appeared to be largely influenced by fluctuations in water level combined with wave action. Water level fluctuations appeared to be mainly determined by wind generated surges and waves, superimposed on tidal patterns (Héquette et al., 1995; Héquette and Barnes, 1990). Although this region is microtidal, with a mean range of just 0.15 m for semidiurnal and monthly tides, these are superimposed on a ca. 0.66 m annual tidal cycle which peaks in late July (Barnhart et al., 2014), corresponding with our intensive observation period. Annual tides therefore likely influence the timing of coastal retreat within the ice-free season in this area. The two periods with the most rapid erosion in

2017 (the 27 days between July 13$^{th}$ to July 30$^{th}$, and the four days between August 11$^{th}$ to August 15$^{th}$) were both associated with strong winds (six-hour moving averages exceeding >40 km h$^{-1}$), both easterly and north-westerly and preceded by relatively high air and water temperatures (Figure 5). Together, these conditions likely enhanced the thermo-abrasional processes undercutting the ice-rich bluff. The high temporal frequency of shoreline position observations is essential to

studying highly episodic erosion processes. For example, ca. 30% (4.2 m) of the 14.5 m of shoreline retreat in the summer of 2017 happened in just four days (August 11$^{th}$ to August 15$^{th}$), indicating discrete storm events can play a major role in the geomorphic development of permafrost shorelines (Farquharson et al., 2018; Solomon et al., 1993). Future work relating coastline change to meteorological and oceanographic factors over short timescales will need to consider the latencies involved between meteorological and oceanographic conditions, undercutting of permafrost cliffs, and planform change as observed

from an aerial perspective.

### 4.3. Rapid coastal erosion as potential threat for the Territorial Parks infrastructure

Coastal change near the Ilutaq – Pauline Cove area influences the stability and evolution of the adjacent gravel spit, which accommodates culturally and historically significant sites, as well as infrastructure essential to the operation of the Qikiqtaruk – Herschel Island Territorial Park. Shoreline change and flooding in recent history has already necessitated the relocation and

raising of several historic buildings at the Ilutaq – Pauline Cove settlement, as well as the relocation of the gravel airstrip essential to the operation of the Park (Olynyk, 2012). However relocation efforts are hindered by the fragility of several buildings, particularly the 'Community House', the oldest building in the Yukon, and it is increasingly difficult to find safer locations for these buildings on the spit (Olynyk, 2012). These historic buildings underpin the site's candidature for UNESCO status (UNESCO, 2004). Erosion of the observed coastal reach exposes the base of the spit to coastal processes, increasing the

risk of changes to the position of the spit itself. Flooding during storm events can isolate the spit from the island (Myers-Smith and Lehtonen, 2016), and such events are projected to become more common in the future (Radosavljevic et al., 2016). Knowledge of coastal processes, particularly patterns of contemporary coastal retreat as a proxy for future patterns, is therefore valuable for informing local management decisions.

### 4.4. Using drones to quantify fine scale coastal erosion dynamics

Drone surveys and photogrammetric analysis are effective tools for measure fine scale erosion dynamics along permafrost coastlines, yielding orthomosaics, inferred shoreline positions (Figures 2 and 3), and surface elevation models (Figure 4). When supplemented by other monitoring of environmental variables (such as wave fields, sea surface temperature, wind strength and direction), drone-acquired observations at fine spatiotemporal resolutions can be related to meteorological and oceanographic observations on supra-annual timescales, providing quantitative insights into erosion processes that vary greatly

in time and space. The temporal resolution of drone surveys can greatly exceed those available by more traditional forms of remote sensing, for example satellite observations or surveys from manned aircraft (Casella et al., 2016; Stow et al., 2004; Whalen et al., 2017), and such observations can be more informative than previously available proxies (such as the apparent

cross sectional area of detached blocks, Barnhart et al., 2014). Such spatial observations could be used to robustly evaluate and refine process-based numerical models of coastal erosion over multiple temporal scales (Barnhart et al., 2014; Casella et al., 2014; Wobus et al., 2011).

Lightweight drones can be deployed at relatively low cost when suitably trained and equipped personnel are on-site. However, the costs of accessing high latitude sites can be substantial, potentially contributing to uneven distributions of monitoring sites (Metcalfe et al., 2018). Surveyable spatial extents depend on the size and the range of the remotely piloted drone, and are also limited by safety and regulatory restrictions. Observations from optical satellites may be better suited for observing change across larger sections of coastline; however, high levels of cloud cover in Arctic regions limits the frequency of successful
optical satellite observations (Hope et al., 2004; Stow et al., 2004). Continuing advances in satellite sensors have increased the spatial resolution and revisit frequency of observations, yet freely available products are currently only available for spatial resolutions of ca. $\geq 10$ m (e.g. Sentinel 2), and finer spatial resolution (< 4 m) products have non-trivial costs for each scene.

In summary, lightweight drone surveys can be suitable when there is a need to accurately measure small changes (e.g. $\leq 0.3$
m) in shoreline positions or elevations over limited extents (e.g. $\leq$ 5-10 km in length). Fine resolution measurements from drone products will be especially useful for isolating the drivers of coastal erosion events, and continued miniaturization of thermal and multispectral cameras for drone platforms will create opportunities to better understand these mechanisms of change. Measurements of surface elevation and consequently volume change can be more informative than simple 2D representations of shoreline position. We generated digital surface models following our drive surveys; however, issues with
insufficient spatial constraint meant that full area coverage was only possible from some of the surveys. If elevation observations are required, care should be taken when conducting drone surveys to ensure that there will be sufficient spatial constraint of the photogrammetric modelling process, even if coastal retreat is faster than expected. For further recommendations on optimising placement of ground control, see Carrivick *et al.* (2016) and James *et al.* (2017).

## 5. Conclusion

We used drones as a highly effective instrument to observe the dynamics of permafrost coastline changes associated with supra-seasonal erosion on Qikiqtaruk – Herschel Island. In 2017, average shoreline retreat was extremely rapid at 14.5 m over 40 days, well in excess of the long-term average of 2.2 m a$^{-1}$ from 1952 to 2017. The volume of material removed was ca. 0.96 m$^3$ m d$^{-1}$ in 2017. Thirty percent of the rapid 2017 shoreline change in (4.1 m retreat) occurred in just four days during one storm event. Block failure was the prevailing mode of erosion, seemingly driven by multiple factors that increase the
susceptibility of the permafrost coastline to thermo-abrasional processes. These rapid erosion events observed on Qikiqtaruk – Herschel Island appear to have been driven by short-term fluctuations in water levels due to meteorological conditions,

possibly superimposed on annual tidal cycles on a longer-term background of relative sea level rise and increasing heat flux from the Mackenzie River discharge and the atmosphere.

We found that lightweight drones and aerial photogrammetry can be cost effective tools to capture short-term coastal erosion dynamics and related shoreline changes along discrete sections of permafrost coasts. At our study site on Qikiqtaruk – Herschel Island further erosion and removal of this coastal reach could threaten the infrastructure of the settlement over the long-term. With the rapid maturation of drone platforms and image-based modelling technologies, these approaches can now be easily deployed at both supra and sub-annual timescales to obtain new insights into coastal erosion and inform management decisions. These approaches are particularly relevant in permafrost coastlines, where erosion can be highly episodic, with long-term rates dominated by short-term events. By combining new methods of observation with long-term records, we can improve predictions of coastal erosion dynamics and subsequent consequences for the management of fragile Arctic coastal ecosystems and cultural sites.

## Video Supplement

A supplementary time-lapse video of the coastal erosion reported here is available at https://doi.org/10.5446/40250.

## Acknowledgements

This work was made possible by support from NERC (ShrubTundra - NE/M016323/1), the loan of GNSS equipment from NERC Geophysical Equipment Facility (GEF:1063 and 1069), the National Geographic Society (Grant CP-061R-17), and the Helmholtz Young Investigators Group 'COPER' (grant #VH-NG-801). This study was supported by funding from the European Union's Horizon 2020 research and innovation programme (Nunataryuk - grant #773421). Drone flight operations were authorised by a Special Flight Operations Certificate granted by Transport Canada. We wish to thank the Qikiqtaruk Territorial Park staff including Richard Gordon, Edward McLeod, Samuel McLeod, Ricky Joe, Paden Lennie, and Shane Goesen, as well as the Yukon Government and Yukon Parks for their permission and support of this research (Permit number Inu-02-16). We also thank the Inuvialuit people for their permission to work on their traditional lands. We thank James Duffy, and anonymous reviewers for insightful feedback that helped us to refine earlier versions of this manuscript.

## Statement of Contribution

Conceptualization, AC, GT, JK and IM-S; Data curation, AC; Formal analysis, AC and GT; Funding acquisition, IM-S, TS, and HL; Investigation, AC, GT, BR, WP and JK; Methodology, AC; Project administration, AC; Supervision, TS, HL and IM-

S; Visualization, AC, GT and WP; Writing – original draft, AC; Writing – review & editing, AC, GT, BR, WP, TS, HL, JK and IM-S.

**Conflicts of Interest**

The authors declare that they have no conflict of interest. The funding sponsors had no role in the design of the study; in the collection, analyses, or interpretation of data; in the writing of the manuscript, or in the decision to publish the results.

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

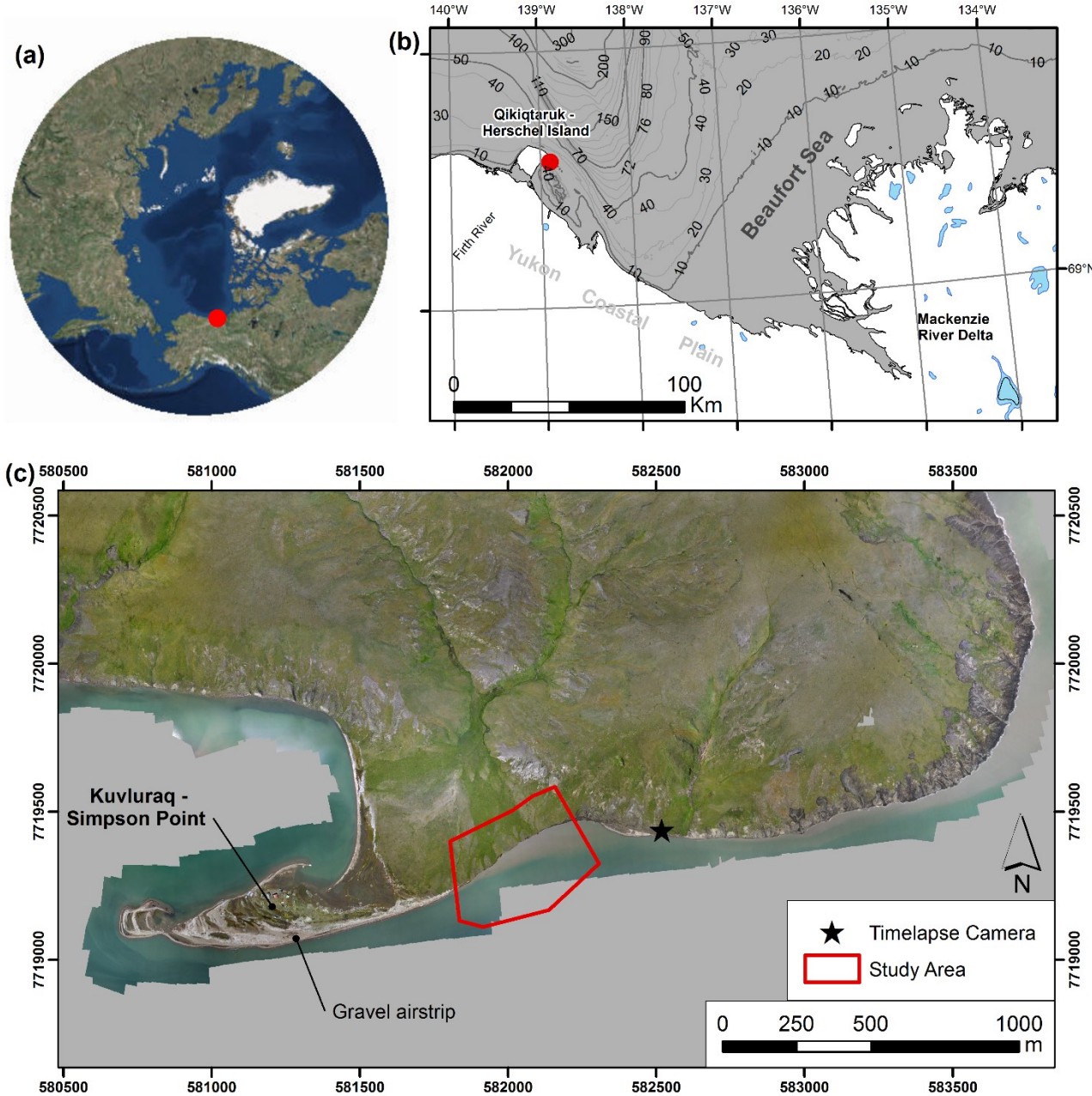

**Figure 1: (a) The location of the study region in the western Canadian Arctic (Basemap from (ESRI et al., 2018), polar stereographic projection), (b) Qikiqtaruk – Herschel Island in the Beaufort Sea (shorelines from (Wessel and Smith, 1996)), and (c) true-colour orthomosaic compiled from ca. 9000 individual images collected by drone survey in August 2017 indicating the location of the 500 m study stretch and the time-lapse camera including viewing direction indicated by the camera symbol used to make the supplementary video (S1) relative to Kuvluraq – Simpson Point.**

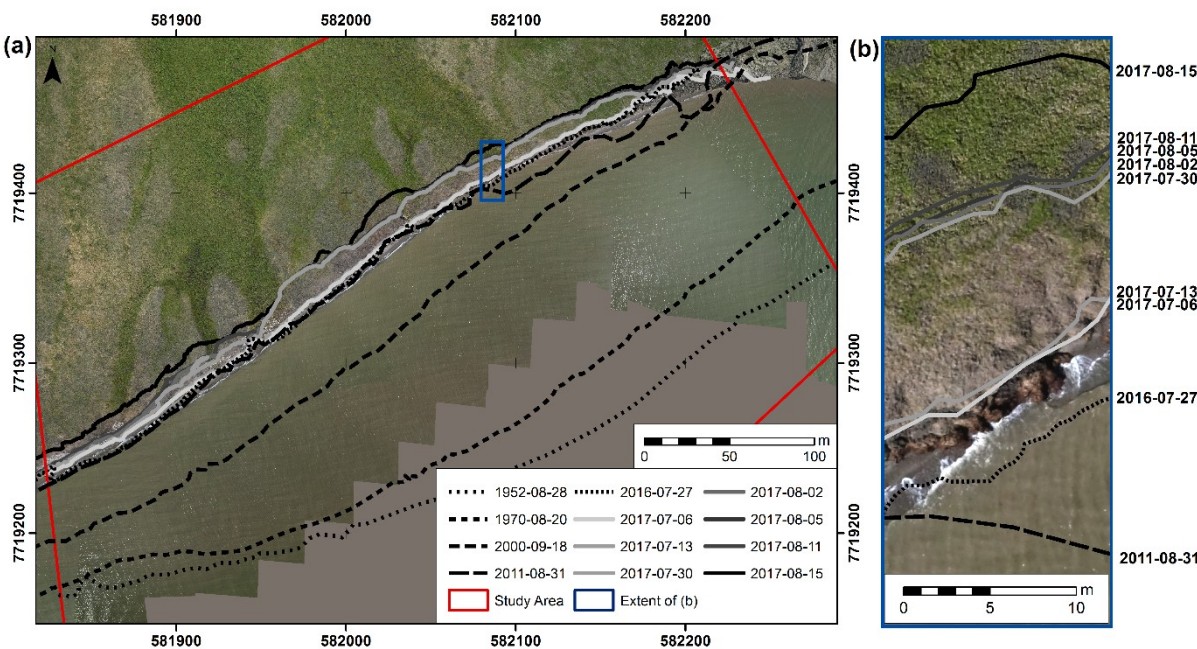

**Figure 2. (a)** Overview of the 500 m study area, illustrating all twelve shoreline positions since 1952 overlaid on the 2017-07-06 orthomosaic. **(b)** Ten-fold magnification in scale, illustrating the episodic nature of the shoreline changes at this location.

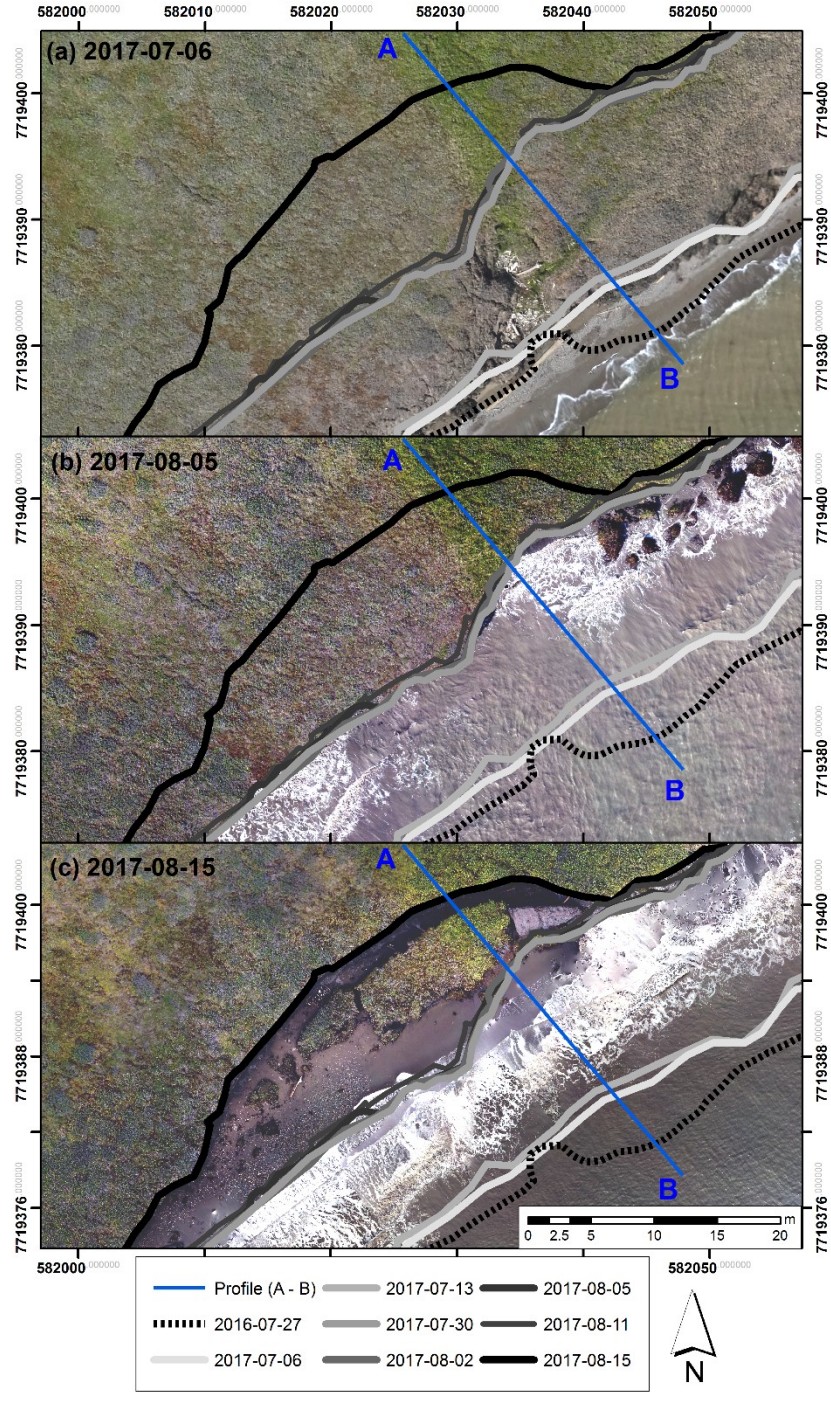

**Figure 3. Shoreline positions between 2016 and 2017 overlaid on three orthomosaics for part of the study reach. The blocks shown in (c) were detached from the bluff, with water moving freely behind during periods of higher water level (see figure S3). Profile A-B indicates the horizontal position of the cross-sectional profiles discussed in section 4.2 and depicted in Figure 4.**

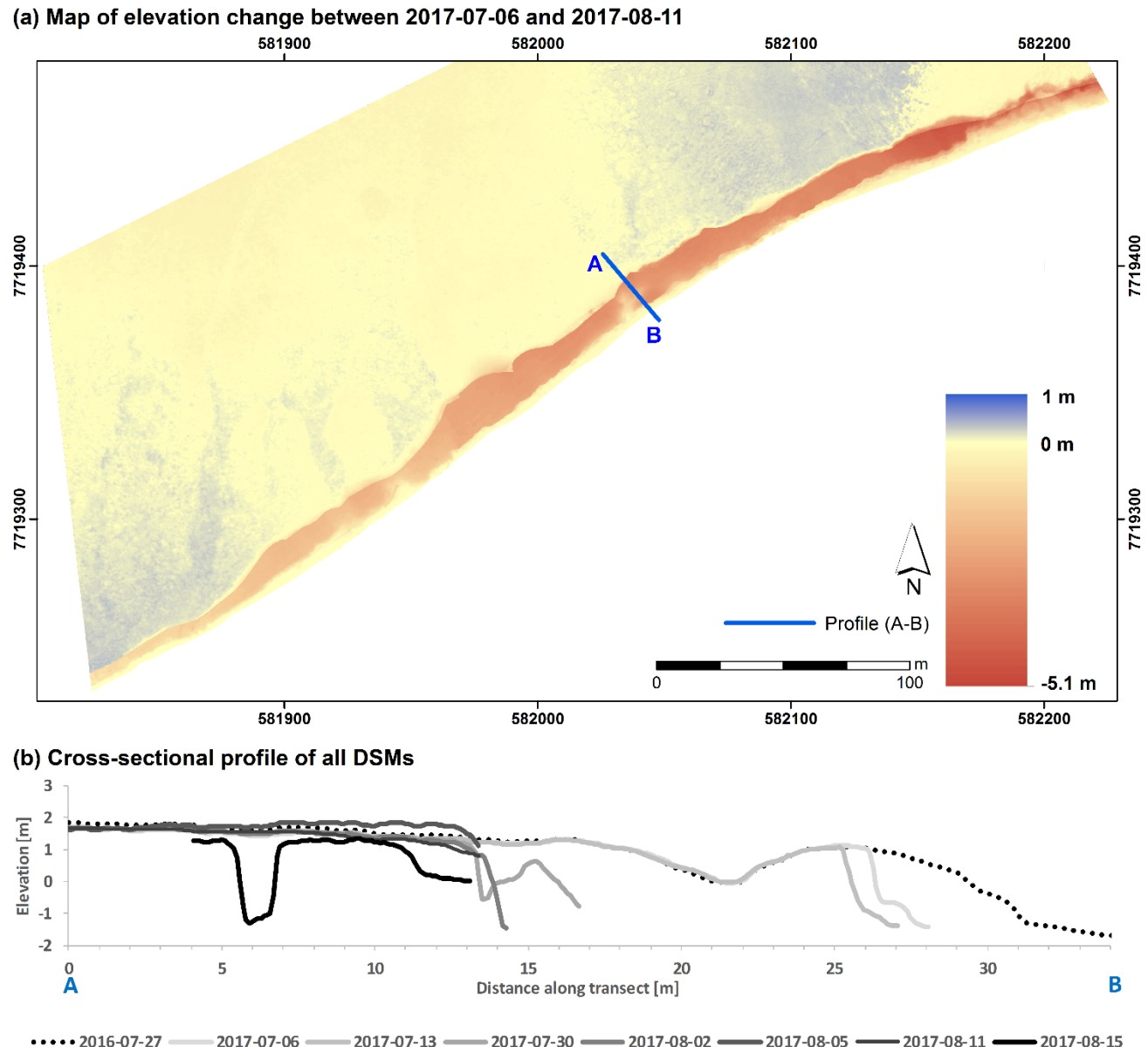

**Figure 4. (a) Map of elevation change in digital surface models between 2017-07-06 and 2017-08-11, illustrating areas of erosion along the coastline with up to -5.1 m change and also some minor (ca. 0.1 m) increases inland that we attribute to vegetation development. (b) Elevation profiles along the A-B transect shown in a and Figure 3, extracted from digital surface models (no vertical exaggeration). Note that two of the latter elevation models (from 2017-08-05 and 2017-08-15) both suffered from datum problems due to insufficient spatial constraint; see discussion for details.**

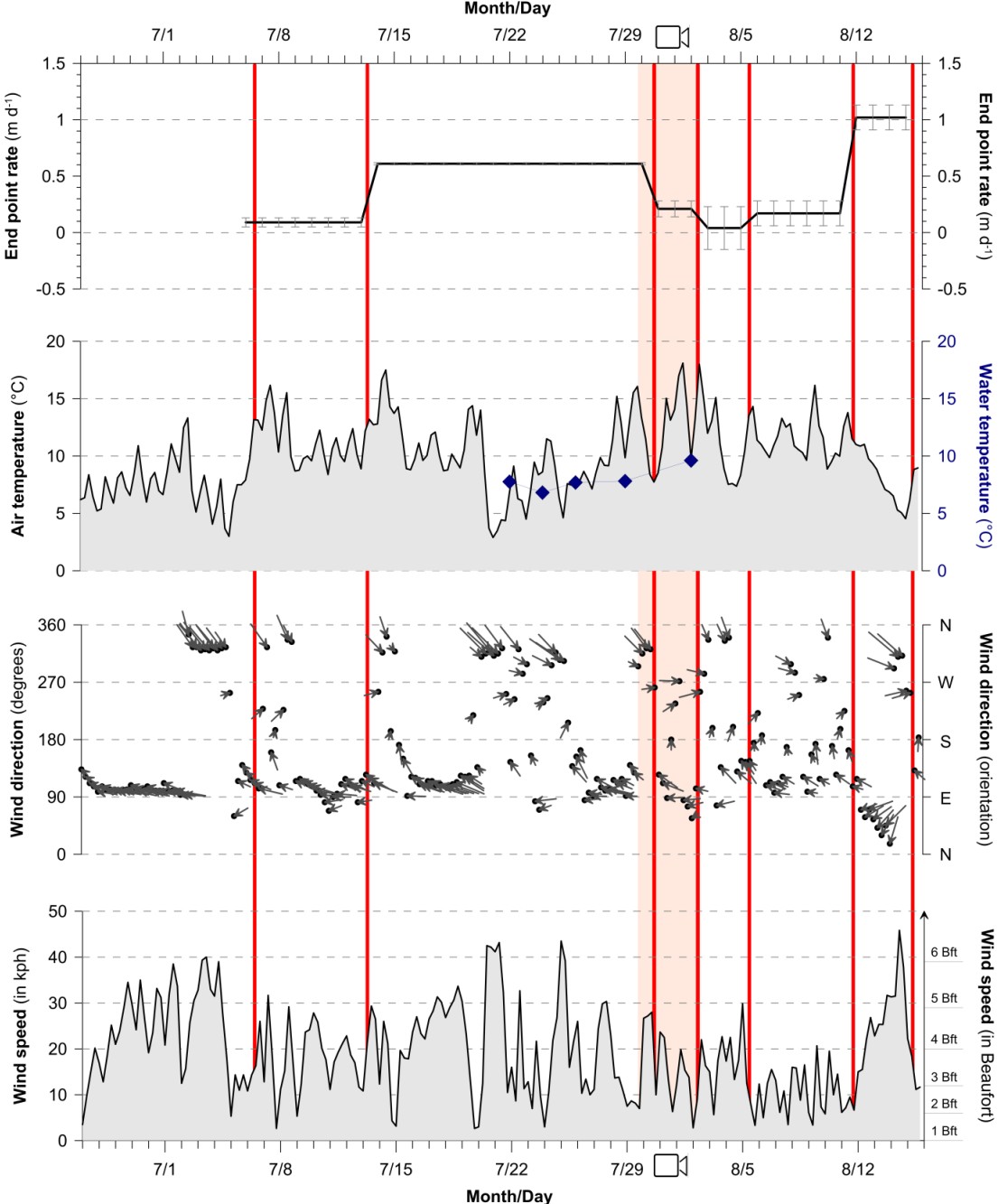

**Figure 5. Shoreline retreat (end point) rates normalised by day for each observation period in 2017, six-hour moving averages of air temperature, wind direction, and wind velocity in July-August 2017 (data from Environment Canada, 2017). Dates and times of aerial surveys are indicated by the red bars, point measurements of sea surface temperature from CTD casts are indicated by blue diamonds (see Figure S3 for full CTD profiles), and the duration of time-lapse survey shown in video S1 is indicated by the red shading.**

**Table 1. Dataset and shoreline position parameters. The approximate times of drone surveys are in local time (UTC -08:00). Total shoreline uncertainty is the root mean square error of (i) image co-registration (root mean square (RMS) error of ground control points (GCP)), (ii) image quality (pixel error), and (ii) shoreline mapping (digitizing) error (from Equation 1).**

| Observation date | Image Type | Altitude [m] | Images [n] | Scale | GCP [n] | Georeferencing RMS Error [m] | | Pixel Error [m] | Digitising Error [m] | Total Shoreline Uncertainty [m] |
|---|---|---|---|---|---|---|---|---|---|---|
| | | | | | | Absolute (NAD83) | Relative | | | |
| 1952-08-28 | Panchromatic aerial photograph | | 1 | 1:70000 | 11 | - | 2.475 | 3.50 | 4.00 | 9.975 |
| 1970-08-20 | Panchromatic aerial photograph | | 1 | 1:12000 | 19 | - | 1.117 | 0.60 | 2.00 | 3.717 |
| 2000-09-18 | Panchromatic Ikonos photograph | | 1 | - | 19 | - | 5.087 | 1.00 | 2.00 | 8.137 |
| 2011-08-31 | Panchromatic GeoEye photograph | | 1 | - | 17 | - | 0.330 | 0.50 | 1.50 | 2.330 |
| 2016-07-27 | RGB orthomosaic (Drone – FX-61) | 120 | 317 | - | 26 | 0.015 | - | 0.03 | 0.15 | 0.195 |
| 2017-07-06 @ 12:20 | RGB orthomosaic (Drone – FX-61) | 120 | 1325 | - | 98 | 0.063 | Base Image | 0.02 | 0.10 | 0.183 |
| 2017-07-13 @ 08:30 | RGB orthomosaic (Drone – FX-61) | 120 | 194 | - | 13 | 0.043 | - | 0.02 | 0.10 | 0.163 |
| 2017-07-30 @ 18:00 | RGB orthomosaic (Drone – Phantom) | 31 | 383 | - | 5 | 0.037 | - | 0.02 | 0.10 | 0.157 |
| 2017-08-02 @ 08:00 | RGB orthomosaic (Drone – Phantom) | 100 | 2040 | - | 22 | 0.021 | - | 0.02 | 0.10 | 0.141 |
| 2017-08-05 @ 11:40 | RGB orthomosaic (Drone – Phantom) | 37 | 336 | - | 6 | 0.443 | - | 0.02 | 0.10 | 0.563 |
| 2017-08-11 @ 17:00 | RGB orthomosaic (Drone – FX-61) | 120 | 8994 | - | 132 | 0.167 | - | 0.04 | 0.15 | 0.307 |
| 2017-08-15 @ 10:20 | RGB orthomosaic (Drone – Phantom) | 42 | 402 | - | 3 | 0.178 | - | 0.02 | 0.10 | 0.298 |

Table 2. Summary of shoreline change for all periods, in terms of net shoreline change and end-point rates. Net shoreline change mean is the distance between the oldest and youngest shorelines, where SD is standard deviation. End-point rate is the net shoreline change normalised by time (years or days, for supra- or sub-annual periods, respectively), where DOA is Dilution of Accuracy (after Equation 2).

| Period | Days | Mean Net Shoreline Change ± SD (m) | Mean End Point Rate ± DOA ($m·a^{-1}$) | ($m·d^{-1}$) |
|---|---|---|---|---|
| Supra-annual periods | | | | |
| 1952-08-28 – 1970-08-20 | 6 567 | 20.7 ± 10.6 | 1.2 ± 0.6 | |
| 1970-08-20 – 2000-09-18 | 10 986 | 69.2 ± 21.9 | 2.3 ± 0.3 | |
| 2000-09-18 – 2011-08-31 | 3 986 | 33.0 ± 11.1 | 3.0 ± 0.8 | |
| 2011-08-31 – 2016-07-27 | 1 791 | 3.5 ± 4.6 | 0.7 ± 0.5 | |
| 2016-07-27 – 2017-07-06 | 344 | 2.9 ± 2.2 | 3.1 ± 0.3 | |
| 1952-08-28 – 2017-08-15 | 23 735 | 143.7 ± 28.4 | 2.2 ± 0.2 | <0.01 ± 0.00 |
| Sub-annual periods | | | | |
| 2017-07-06 – 2017-07-13 | 7 | 0.5 ± 0.5 | | 0.09 ± 0.04 |
| 2017-07-13 – 2017-07-30 | 17 | 7.4 ± 5.6 | | 0.61 ± 0.01 |
| 2017-07-30 – 2017-08-02 | 3 | 0.6 ± 1.1 | | 0.21 ± 0.07 |
| 2017-08-02 – 2017-08-05 | 3 | 0.1 ± 0.4 | | 0.04 ± 0.19 |
| 2017-08-05 – 2017-08-11 | 6 | 1.0 ± 0.4 | | 0.17 ± 0.11 |
| 2017-08-11 – 2017-08-15 | 4 | 4.1 ± 1.1 | | 1.02 ± 0.11 |
| 2017-07-06 – 2017-08-15 | 40 | 14.5 ± 3.2 | | 0.36 ± 0.01 |