# Peer review of "Rapid retreat of permafrost coastline observed with aerial drone photogrammetry"

_The Cryosphere, 2018_

## Referee Comment (RC1) · Anonymous Referee #1 · 9 Jan 2019

General comments The manuscript presents investigations of short-term coastal dynamics at a very rapidly retreating coastline using UAVs (drones) combined with data on long-term coastal dynamics of the same section according to satellite and aerial images. Although using multitemporal imagery analysis for coastal retreat measurements is common practice, and Herschel Island is a relatively well studied area in terms of coastal dynamics, the authors made the first attempt to provide very high temporal resolution observations of coastal erosion, including intra-seasonal dynamics presented by short-term periods (3-7 days during the summer of 2017). This is the principal novelty of the study, which gave new insights into mechanisms and rate variability of coastal erosion and proved again its episodic nature, when a coastal segments can retreat by several meters in a few days during one storm. In this way, the investigated

coastal segment gave a unique opportunity for such detailed analysis, as the rates of retreat in 2017 were unprecedented. Another strong point of the manuscript is the well-described methodology, giving an example of using drones for coastal dynamics monitoring, which is already popular and will surely become one of the main tools in coastal investigations in the years to come. We would advise to reduce some general comments about the evident benefits of using drones and focus on giving more technical details that can be further used for elaboration of technologic standards (flight heights, required number of ground control markers, etc. - see in Specific comments below). Overall, the manuscript is a high quality study, with valid and appropriate methods, new trustful results supporting the discussion, fluent and precise language, well-readable figures and abundant supplementary material. The discussion can be re-grouped and some sections of it shortened (see below), however, this does not hinder the general good impression of the paper.

Specific comments Abstract The abstract might be shortened, omitting information on the Kuvluraq – Simpson Point gravel spit, which is mentioned in the text shortly. The objectives can be shortened. The phrases: Lines 28-30 ("We found drone surveys analysed with image-based modelling yield fine-grain and accurately geolocated observations that are highly suitable to observe intra-seasonal erosion dynamics") and Lines 33 Page 1 - 2 Page 2 (We conclude that the data available from drones is an effective tool to understand better the mechanistic short-term controls on coastal erosion dynamics and thus long-term coastline change, and has strong potential to support local management decisions regarding coastal settlements in rapidly changing Arctic landscapes") are somewhat repetitive, and one of them can be omitted Introduction Page 2, Line 8 - "Coastal erosion is prevalent along the Western North American Arctic coastline and Eastern Siberia" - what about significant erosion rates in Western Siberia and in Western Russia along the Pechora Sea coasts? (Vasiliev et al., 2005, Kritsuk et al., 2014, Ogorodov et al., 2016, Novikova et al., 2018) Kritsuk, L.N.; Dubrovin, V.A.; Yastreba, N.V. Some results of integrated study of the Kara Sea coastal dynamics in the Marre-Sale meteorological station area, with the use of GIS technologies. Earth's Cryosphere, 2014, 4, 59–69. http://www.izdatgeo.ru/pdf/earth_cryo/2014-4/52_eng.pdf Vasiliev, A.; Kanevskiy, M.; Cherkashov, G.; Vanshtein, B. Coastal dynamics at the Barents and Kara Sea key sites. Geo-Mar. Lett. 2005, 25, 110–120. https://link.springer.com/article/10.1007%2Fs00367-004-0192-z Ogorodov, S., Baranskaya, A., Belova, N., Kamalov, A., Kuznetsov, D., Overduin, P., Shabanova, N., and Vergun, A. (2016). Coastal dynamics of the Pechora and Kara seas under changing climatic conditions and human disturbances. GEOGRAPHY, ENVIRONMENT, SUSTAINABILITY, 9(3):53–73. Novikova A., Belova N., Baranskaya A., Aleksyutina D., Maslakov A., Zelenin E., Shabanova N., and Ogorodov S., Dynamics of permafrost coasts of Baydaratskaya bay (Kara sea) based on multi-temporal remote sensing data, Remote Sensing 10 (2018), no. 1481 Is there direct evidence that coastal erosion prevails over accumulation in the mentioned regions? Is the sum of erosional segments overall longer than the sum of accumulative segments? If not, would be better to rephrase, e.g., "rates of coastal erosion are considerable", or "the fastest coastal erosion was documented..." or "coastal erosion has high rates" Methods Section 3.1. Page 4, Line 32 Artificial ground control markers were deployed along the shoreline and precisely geolocated to an absolute accuracy of centimetres using global navigation satellite system (GNSS) equipment (Leica Geosystems). If it is possible it would be interesting to mention how the used number of markers was chosen, and how many markers are sufficient, depending on the study site characteristics? Page 6, Lines 19-20: Total shoreline uncertainties were calculated as the sum of georeferencing, pixel and digitising errors (Radosavljevic et al., 20 2016; Río and Gracia, 2013), and survey parameters and shoreline errors are given in Table 1. Why aren't the total uncertainties calculated as the root mean square error (square root of the sum of the squares of independent errors)? Page 6, Lines 13-14 "Shoreline digitising errors were derived from the estimated accuracy of operator vegetation edge detection, informed by reference to finer grain aerial imagery" - not sure I understood well from this fragment how exactly the digitising errors were calculated. Was it by comparison of digitising by different operators? Why are they the same for all drone images from 2017? Page

7, Lines 8-9 "To inform qualitative interpretation of the erosion dynamics at this location, a time-lapse camera was installed at the location indicated on Figure 1 between 2017-07-29 and 2017-08-03." - this goes to section 3.1 (it can be called "Fieldwork and UAV image acquisition") or to section 3.2. Anyway, it's neither meteorological nor oceanographic data Results After the drone surveys, DEMs were built, from which profiles are provided in Figure 4. Why are there no calculations of volumes of the material eroded in 2016-2017? Would be good to provide pictures in 3D. The authors faced some problems with the destroyed ground control markers; however, there could be some conclusions on the volume with smaller accuracy, and/or for the periods between surveys with good quality referencing only Page 7, Lines 12-15. Are you speaking about average values of retreat for the 500-m coastal segment? What was the spatial variability of coastal erosion? If 14.5 $\pm$ 3.2 m was an average distance of retreat in 2017, were there locations with greater or smaller retreat, and what were the extremes? You are showing that coastal retreat was episodic in time, and saying it was also episodic in space - could you highlight examples in the text? Page 8, Lines 23-25 "A timelapse video illustrating the erosion at this coastline over five days from the location marked in Figure 1 is presented in video 25 S1" - could you please describe here very briefly what exactly the video shows? Discussion The grouping of the Discussion is not always logical and needs to be revised. One of the suggestions is to move Section 5.1 to the end of the discussion. Otherwise, the introductory paragraph (page 8, Lines 27-31, Page 9 Lines 1-2) should be put after it. According to our opinion, Section 5.1 is too long and contains much obvious information that can be omitted without harm to the general content. Part of this is somewhat repetitive to the Introduction, other information can be moved to the Introduction. Lines 10-15 belong to other sections of the Discussion, e.g., Section 5.3. Page 10, Lines 9-10 "Fine spatial grain measurements from drone products are especially useful for isolating the drivers of coastal erosion events" - would be good to provide exact examples from the study site where you could isolate the drivers of separate coastal erosion events you are describing Section 5.2 There is no discussion on spatial variability of coastal erosion

rates during short periods (e.g., 2017) and its reasons. Would be good to add it. Could you state precisely, what is the main short-term driver, according to your findings? Is it the wind speed? Might be a good idea to try to build a quantitative correlation between the wind speed and the erosion rates during the investigated period? Page 11, Lines 9-10. Is there any quantitative data on sea-level fluctuations during the observations? Section 5.3. Would be good to provide some brief information on hydrometeorological conditions of the past years and discuss why 2017 was characterized by such dramatic retreat rates compared with previous years . You are speaking about the ice-free period increase, temperature growth, increased wave height, war water discharge, but all of these factors were already present in 2016, 2015, etc. - what is your opinion of why coastal erosion accelerated so much namely in 2017? The name of Section 5.4 does not match its content. This section describes coastal erosion at Herschel Island in the context of long-term erosion rates at different locations around the Arctic, rather than short-term coastal erosion in the context of long-term observations

Technical corrections Page 1 Line 31 change to " Over a single four-day period" Line 32 " exceeded 1 ± 0.1 m d -1" - Please be consistent with number formats, and the number of decimals. If you previously reported the number of "2.2 ± 0.2 m a-1", you should provide this number as "1.0 ± 0.1 m d -1" Page 2 Line 11 - and affect? Line 20 - "improved understanding is required" Page 3 Line 6 - "repeated drone surveys" Lines 5-11. I would advice to use the present tense, rather than the past tense (e.g., "In this study, we use...") Lines 10-12 "We demonstrated that lightweight drones and aerial photogrammetry can be cost effective tools to capture short-term coastal erosion dynamics and related shoreline changes along discrete sections of permafrost coasts." - This goes to the conclusions Figure 1c - remove "Text" from the top right side of the map? Line 17 - please add a reference for Figure 1a Line 20 - "the mean annual air temperature is..."; "the mean annual precipitation is..." Line 22 - "between 2000 and 2011" or "in 2000-2011" Line 24 - delete "in this region" Line 28 - northwesterly and easterly winds; "they exert..." "and with easterly winds facilitating the transport of warm water from the Mackenzie River to Qikiqtaruk Herschel Island" - unfinished

Interactive
comment

phrase? Facilitate? Page 4 Line 1 - sea-level Page 5 Line 14 - Processing parameters are reported... Page 6 Line 20 " Río and Gracia, 2013), and survey parameters" - replace by " Río and Gracia, 2013); survey parameters" "and survey parameters and shoreline errors are given in Table 1" - this reference goes to section 3.1 (regarding the survey parameters); the reference to Table 1 in the context of shoreline position errors is repetitive with Lines 15-17 Line 25 - delete "calculated" Page 7 Line 5 - Figure 5 should be mentioned after the reference in the text to Figures 2, 3 and 4 Line 12: " by a net total of 143.7 $\pm$ 28.4 m" - is it an average value for the whole segment? Line 16 "shoreline retreat was 14.5 $\pm$ 3.2 m, an average rate of 36 cm per day." - replace by "the shoreline retreated by 14.5 $\pm$ 3.2, with an average rate of" Line 17 - the shoreline positionS Line 18 - meant that THE shorelines Lines 19-20 "Coastline retreat was highly episodic in time and space, occurring primarily over two periods" - repetitive, replace by "Coastline retreat primarily occurred over two periods" Line 21: " There was minimal change in coastline position DURING SIX DAYS between August 5th and August 11th Line 25: " a 13-month period" Line 26: " in Figure 4, sampled across the A-B-transect indicated on Figure 3." - replace " in Figure 4; they were sampled across the A-B-transect indicated in Figure 3" Page 8 Line 3 - from three to ten days Line 4 - and their speed reached up to... Line 5 "For zero to three days prior to the 1st 5 2017 survey (on 2017-07-06)" replace by " For zero to three days prior to the same survey" Line 11 - of very strong winds Line 16 - and facilitate further undercutting Line 18 - and the wind speed was low Lines 20-21 These meteorological conditions resulted in large waves and undercutting - sounds more logically Line 29 - where retreat rates typically range Line 30 - between 0 and 2 m Page 9 Lines 13-14 - "In this case, however, for the total 17.4 m of shoreline retreat between 2016 and 2017 reported here to remain consistent with the long-term average of 2.2 m a -1 , no further erosion of this reach would need to occur 15 for more than seven years" - rephrase: In this case, however, to remain consistent with the long-term average of 2.2 m a -1 , no further erosion of this reach would need to occur for more than seven years after the retreat of 17.4 m in 2016-2017. Page 11 Lines 2-3 " Further factors facilitating rapid erosion at this coastal

reach ARE the high ice content (ca. 40% Obu et al., 2016) and the low relief" Line 10 - Although this region is microtidal - "although the studied region is microtidal" Page 11, Lines 13-14 - "Winds exert substantial control over local sea levels, with north-westerly winds driving a positive storm surge and easterly winds driving a negative storm surge (Héquette et al., 1995; Héquette and Barnes, 1990)." - repetitive; already appeared in the Introduction Page 12 Line 11 - on Bykovsky Peninsula? Figures: Figure 1c - remove "Text" from the top right side of the map? Figure 2. What is the image at the background? Figures 2, 3 and 4. Would be better readable if you used different colours for coastlines of different time periods instead of shades of grey and black

Please also note the supplement to this comment:
https://www.the-cryosphere-discuss.net/tc-2018-234/tc-2018-234-RC1-supplement.pdf

---

## Referee Comment (RC2) · Anonymous Referee #2 · 1 Feb 2019

Cunliffe et al. present a case study for an eroding permafrost coastline along the Canadian Beaufort Sea Coast using historic photos, satellite images, and airborne images acquired from a UAV. The imagery ranged in spatial resolution from 3.5 m to 0.02 m and consisted of images acquired between 1952 and 2017 and focused on a 500 m segment of coastline. It appears that the historic imagery was already published previously (could be better clarified in the paper) and the novelty of this paper was the high temporal image acquisition using UAVs in 2017. Seven UAV surveys were used to create high spatial resolution orthophotos and digital surface models to assess coastal change rates on the order of days to weeks during July and August of 2017.

The paper is well written and organized. The study design and presentation of results are clear but need to be improved. In particular, the mismatch in image spatial reso-

lution and temporal observations require further consideration in the paper. A number of suggested edits and revisions are provided below to help refine the paper to make it suitable for publication in The Cryosphere.

General Comments

- The comparison between decadal-scale erosion rates from images with a spatial resolution that ranged from 0.5 m to 3.5 m with coastal change positions determined from images with a spatial resolution of 0.02 to 0.04 m requires further validation. This is particularly important given the assertion that erosion rates in 2017 was 14.5 m/yr compared to a long-term average rate of 2.2 m/yr, or as stated in the abstract more than six times faster. The authors need to include a suitable image from 2017 or 2018 at a resolution that is more in line with image resolutions available historically to demonstrate that the increased resolution of the UAV imagery is not responsible for the measured increase in erosion, simply due to being able to better detect the feature of interest. Doing a quick survey of images available from DigitalGlobe shows that there are some potential options available for the study site in 2017 and 2018 that could provide this necessary check.

- On line 30 of the abstract the authors report that in 2017 mean coastal retreat was 14.5 m/yr. However, in table 2 it appears that there were only 40 days of erosion analyzed during this period. It appears that the 14.5 m of erosion refers to the magnitude of shoreline change and not an annual rate. This critical point needs to be better clarified and the mismatch in temporal periods among observation periods given more careful consideration. One consideration could be that the image acquired on 2016-07-27 be compared with the image acquired on 2017-08-15 to determine the most recent annual erosion rate instead of using the 2017-07-06 for this. Reporting it in this manner and then using the UAV image acquisitions within this latter annual-scale period to assess event driven erosion patterns and controls might make for a cleaner analysis and presentation of results.

- Considering that the historic remote sensing data was apparently previously published (is this what previously analysed refers to on line 23 page 5) the authors need to enhance their methodology and presentation of the imagery acquired with the UAV surveys. The authors should provide information on the altitude of the UAV during image acquisition, the orientation of the flight paths relative to the coastline, why they recommend using front lap and side of 10 and 20 respectively while only using 5 and 10 respectively, the number of ground control points established in the field, and why the authors did not constrain their orthophotos and digital surface models using ground check points when this method is recommended in the literature. All of this should be correctable and is not seen as a major sticking point. The authors are also encouraged to maximize the use of their UAV data by analyzing the digital surface models constructed in Agisoft. Currently this assessment consists of four sentences in the results section. The authors mention that erosion occurring after the fourth UAV survey prevented proper construction of digital surface models in the latter efforts. However, the digital surface model data acquired during the first surveys combined with the shoreline positions digitized from the latter time period orthophoto mosaics should provide sufficient information to add this element to the paper.

Specific Edits and Questions

- Replace the use of grain with resolution throughout the paper

- Consider changing the use of drone to UAV throughout the paper

- Equation 1 seems to be incomplete according to variables presented in Table 1 to determine shoreline change uncertainty. Check this.

- Was the CTD data acquired in 2015 or during the study period in 2017. Check line 7 on page 7. If from 2015 how is it relevant to this study?

- Specify whether the time lapse camera in operation for 4 days imaged the study coastline during the observation period.

[Figure]

- Change cm per day on line 16, page 7 to m per day

- Please explain the significance of the linear regression method being more conservative than the end point method as reported on lines 23-25, page 12

- Adding field photos of the study coast would add useful information to the paper and provide a context for understanding the permafrost characteristics at the site

---

## Author Response (AR1)

**Rapid retreat of permafrost coastline observed with aerial drone photogrammetry - Response to Referees**

**Referee #1**

General comments
5   The manuscript presents investigations of short-term coastal dynamics at a very rapidly retreating coastline using UAVs (drones) combined with data on long-term coastal dynamics of the same section according to satellite and aerial images. Although using multitemporal imagery analysis for coastal retreat measurements is common practice, and Herschel Island is a relatively well studied area in terms of coastal dynamics, the authors made the first attempt to provide very high temporal resolution
10   observations of coastal erosion, including intra-seasonal dynamics presented by short-term periods (3-7 days during the summer of 2017). This is the principal novelty of the study, which gave new insights into mechanisms and rate variability of coastal erosion and proved again its episodic nature, when a coastal segments can retreat by several meters in a few days during one storm. In this way, the investigated coastal segment gave a unique opportunity for such detailed analysis, as the rates of
15   retreat in 2017 were unprecedented. Another strong point of the manuscript is the well described methodology, giving an example of using drones for coastal dynamics monitoring, which is already popular and will surely become one of the main tools in coastal investigations in the years to come. We would advise to reduce some general comments about the evident benefits of using drones and focus on giving more technical details that can be further used for elaboration of technologic standards (flight
20   heights, required number of ground control markers, etc. - see in Specific comments below). Overall, the manuscript is a high quality study, with valid and appropriate methods, new trustful results supporting the discussion, fluent and precise language, well-readable figures and abundant supplementary material. The discussion can be re-grouped and some sections of it shortened (see below), however, this does not hinder the general good impression of the paper. Please also note the
25   supplement to this comment: https://www.the-cryosphere-discuss.net/tc-2018-234/tc-2018-234-RC1-supplement.pdf

**Author response: We thank Referee 1 for their very positive appraisal of our manuscript, and for their constructive suggestions. We have revised our manuscript in light of this feedback (responses to specific points of feedback provided**
30   **below), and hope that the Referee will agree the manuscript is now greatly improved as a result.**

Specific comments
Abstract
35   The abstract might be shortened, omitting information on the Kuvluraq – Simpson Point gravel spit, which is mentioned in the text shortly. The objectives can be shortened. The phrases: Lines 28-30 ("We found drone surveys analysed with image-based modelling yield fine-grain and accurately geolocated observations that are highly suitable to observe intra-seasonal erosion dynamics") and Lines 33 Page 1 - 2 Page 2 (We conclude that the data available from drones is an effective tool to
40   understand better the mechanistic short-term controls on coastal erosion dynamics and thus long-term

coastline change, and has strong potential to support local management decisions regarding coastal settlements in rapidly changing Arctic landscapes") are somewhat repetitive, and one of them can be omitted

5   **Author response: Thank you for these helpful suggestions, we have revised our abstract, shortening it by ca. 25%.**

Introduction
Page 2, Line 8 - "Coastal erosion is prevalent along the Western North American Arctic coastline and
10   Eastern Siberia" - what about significant erosion rates in Western Siberia and in Western Russia along the Pechora Sea coasts? (Vasiliev et al., 2005, Kritsuk et al., 2014, Ogorodov et al., 2016, Novikova et al., 2018). Is there direct evidence that coastal erosion prevails over accumulation in the mentioned regions? Is the sum of erosional segments overall longer than the sum of accumulative segments? If not, would be better to rephrase, e.g., "rates of coastal erosion are considerable", or "the fastest
15   coastal erosion was documented..." or "coastal erosion has high rates"

**Author response: Thank you for highlighting this additional literature. We have refined the text here which now reads "Coastal erosion is prevalent along the Western North American Arctic coastlines, and all of Siberia, and is one of the major key processes degrading permafrost (Lantuit et al., 2012)." We have now read these papers, and where**
20   **appropriate, have integrated them into our manuscript to strengthen the linkages with this body of knowledge on this topic. These papers show coastline segments characterized by erosion and accumulation with some coastlines characterized by a majority of accumulative segments (e.g. Novikova et al., 2018). Yet, the papers also showed that many of these segments have now become erosive over the past few years, reflecting a shift from accumulative to erosive coastlines also observed in the study region (Irrgang et al., 2018).**

Methods Section 3.1.
Page 4, Line 32 Artificial ground control markers were deployed along the shoreline and precisely geolocated to an absolute accuracy of centimetres using global navigation satellite system (GNSS)
30   equipment (Leica Geosystems). If it is possible it would be interesting to mention how the used number of markers was chosen, and how many markers are sufficient, depending on the study site characteristics?

**Author response: We would recommend 13 ground control markers as the ideal number per photogrammetric**
35   **reconstruction, which might cover a coastline reach of up to a few km in length. Ten markers would be used to constrain the bundle adjustment and three to evaluate the photogrammetric reconstruction (as recommended by James et al. 2017).**

Page 6, Lines 19-20: Total shoreline uncertainties were calculated as the sum of georeferencing, pixel and digitising errors (Radosavljevic et al., 2016; Río and Gracia, 2013), and survey parameters and shoreline errors are given in Table 1. Why aren't the total uncertainties calculated as the root mean square error (square root of the sum of the squares of independent errors)?

**Author response: We used additive rather than quadratic error propagation because the pixel, georegistration, and digitising errors are not independent, with high pixel error (due to coarse resolution) resulting in higher registration and digitising errors (Table 1). Consequently, it is more appropriate to use the more conservative additive approach to error propogation (as also used by Radosavljevic et a;. 2016). In any case, there is minimal difference between the total**
10 **uncertainties in either shoreline position or end point rates between additive or quadratic approaches to error propagation. We have added explanation for our choice here in the methods section.**

Page 6, Lines 13-14 "Shoreline digitising errors were derived from the estimated accuracy of operator
15 vegetation edge detection, informed by reference to finer grain aerial imagery" - not sure I understood well from this fragment how exactly the digitising errors were calculated. Was it by comparison of digitising by different operators? Why are they the same for all drone images from 2017?

**Author response: The digitising error was estimated by an experienced operator, and we believe that the estimated**
20 **error terms are conservative in relation to the spatial resolution of the classified images (e.g. errors of 0.10 m when spatial grain is 0.02 m). We believe that using multi-operator comparisons to estimate digitisation error can sometimes be incomplete characterisation of error, as they do not evaluate against a true position and can fail to account for all operators making the same mistakes. We have revised the manuscript to explain this approach more clearly; it now reads "Shoreline digitising errors were estimated by the GIS operator, and ranged between 0.1 m and 4.0 m depending**
25 **on image spatial resolution (Table 1)."**

Page 7, Lines 8-9 "To inform qualitative interpretation of the erosion dynamics at this location, a time-lapse camera was installed at the location indicated on Figure 1 between 2017-07-29 and 2017-08-03."
30 - this goes to section 3.1 (it can be called "Fieldwork and UAV image acquisition") or to section 3.2. Anyway, it's neither meteorological nor oceanographic data

**Author response: We have moved this information as suggested by the reviewer (to section 2.1 in the revised manuscript).**

Results After the drone surveys, DEMs were built, from which profiles are provided in Figure 4. Why are there no calculations of volumes of the material eroded in 2016-2017? Would be good to provide pictures in 3D. The authors faced some problems with the destroyed ground control markers; however,

there could be some conclusions on the volume with smaller accuracy, and/or for the periods between surveys with good quality referencing only.

**Author response: Following this constructive suggestion, we have extended our analysis of surface elevation changes to including estimates of volumes of erosion. We have updated the methods, results and discussion sections of the manuscript to reflect these changes.**

Page 7, Lines 12-15. Are you speaking about average values of retreat for the 500-m coastal segment? What was the spatial variability of coastal erosion? If 14.5 ± 3.2 m was an average distance of retreat in 2017, were there locations with greater or smaller retreat, and what were the extremes? You are showing that coastal retreat was episodic in time, and saying it was also episodic in space - could you highlight examples in the text?

**Author response: Yes, this text described average values across the 500 m segment. We have updated the text to include more description of the spatial variation in shoreline change, including the maximum (22 m) and minimum (6 m) retreat rates observed over this 40-day period.**

Page 8, Lines 23-25 "A timelapse video illustrating the erosion at this coastline over five days from the location marked in Figure 1 is presented in video S1" - could you please describe here very briefly what exactly the video shows?

**Author response: As recommended, we have expanded the text here to describe the contents of this video and added a camera symbol and viewing angle in Figure 1 (c).**

Discussion The grouping of the Discussion is not always logical and needs to be revised. One of the suggestions is to move Section 5.1 to the end of the discussion. Otherwise, the introductory paragraph (page 8, Lines 27-31, Page 9 Lines 1-2) should be put after it. According to our opinion, Section 5.1 is too long and contains much obvious information that can be omitted without harm to the general content. Part of this is somewhat repetitive to the Introduction, other information can be moved to the Introduction. Lines 10-15 belong to other sections of the Discussion, e.g., Section 5.3.

**Author response: Thank you for highlighting this area for improvement. We have extensively revised the structure and content of our discussion in line with these constructive recommendations. We now have four sections, entitled: 4.1.**

Rapid shoreline change, 4.2. Drivers of rapid shoreline change, 4.3. Rapid coastal erosion as potential threat for the Territorial Parks infrastructure, and 4.4. Using drones to quantify fine scale coastal erosion dynamics.

5  Page 10, Lines 9-10 "Fine spatial grain measurements from drone products are especially useful for isolating the drivers of coastal erosion events" - would be good to provide exact examples from the study site where you could isolate the drivers of separate coastal erosion events you are describing Section 5.2 There is no discussion on spatial variability of coastal erosion rates during short periods (e.g., 2017) and its reasons. Would be good to add it. Could you state precisely, what is the main

10  short-term driver, according to your findings? Is it the wind speed? Might be a good idea to try to build a quantitative correlation between the wind speed and the erosion rates during the investigated period?

**Author response: Thank you for this question of attribution. In this study we unfortunately do not have sufficient continuous ancillary observations of key parameters to robustly extend this analysis further (i.e. sea level, sea surface**
15  **temperature, wave direction and energy). We know that wind direction matters for erosion rates at this locale, we expect a non-linear relationship between wind speed and erosion (as found by Vasiliev et al., 2005), and we know that there are latencies between erosion (especially undercutting) and shoreline change as observed from a nadir perspective. This complexity in process interactions is confounded by there being just six short-term periods of coastline change, ranging from 3 to 17 days duration. Consequently, we feel that further attribution analysis is outside the scope**
20  **of this manuscript, but we strongly agree that it would be valuable for future work in this area to use these tools we demonstrate to examine this question of attribution in greater detail. We have revised the text of our discussion for greater clarity on this point.**

25  Page 11,
Lines 9-10. Is there any quantitative data on sea-level fluctuations during the observations?

**Author response: Unfortunately, there are no quantitative observations of sea level fluctuations available at this location, and so consequently, we used '…appeared to be…' to indicate the qualitative nature of our observations. A**
30  **tide gauge has now been installed since 2018 so we are hoping to record this parameter in the future.**

Section 5.3. Would be good to provide some brief information on hydrometeorological conditions of the past years and discuss why 2017 was characterized by such dramatic retreat rates compared with
35  previous years. You are speaking about the ice-free period increase, temperature growth, increased wave height, war water discharge, but all of these factors were already present in 2016, 2015, etc. - what is your opinion of why coastal erosion accelerated so much namely in 2017?

**Author response: Thank you for this question. Unfortunately, while we were able to detect substantial changes over**
40  **our observation period and discuss our findings in the context of broader regional hydrometeorological conditions, the limited hydrometeorological observation available at this specific location limits our ability to attribute these rapid**

**changes in 2017 to specific drivers.** We have looked at the start of the open water season, inferred from MODIS observations (Nasa WorldView, https://worldview.earthdata.nasa.gov/). This suggests that the sea ice may have moved out earlier than normal in the years immediately preceding 2017 which could have helped to condition this permafrost cliff (Figure R1 below, now added to the Supplementary Information). However, this inference is highly speculative. Although it would be possible to extract data on sea ice coverage (e.g. from the National Snow and Ice Data Centre) and meteorological conditions (e.g. from the Government of Canada's observations), these records are temporally patchy and do not encompass important parameters such as sea level (the first tide gauge in the area was temporarily installed in July-August 2018) and wave regimes (the first Acoustic current doppler profiler was temporarily installed in 2018 for a period of <1 month and the closest NOAA buoy is more than 200 km away, https://www.ndbc.noaa.gov/station_page.php?station=48021). We are highly doubtful that intensive analysis of *available* hydrometeorological data would result in an explanation of the rapid change observed in 2017.

The name of Section 5.4 does not match its content. This section describes coastal erosion at Herschel Island in the context of long-term erosion rates at different locations around the Arctic, rather than short-term coastal erosion in the context of long-term observations

**Author response: We have integrated this material into section '4.2. Drivers of rapid shoreline change', and believe that the revised text is much more coherent.**

Technical corrections
Line 31 change to " Over a single four-day period"

**Author response: Correction implemented.**

Line 32 " exceeded 1 _ 0.1 m d -1" - Please be consistent with number formats, and the number of decimals. If you previously reported the number of "2.2 _ 0.2 m a-1", you should provide this number as "1.0 _ 0.1 m d -1"

**Author response: Correction implemented.**

Line 11 - and affect?

**Author response: Correction implemented.**

Line 20 - "improved understanding is required"

**Author response: Correction implemented.**

Line 6 - "repeated drone surveys"

**Author response: Correction implemented.**

Lines 5-11. I would advice to use the present tense, rather than the past tense (e.g., "In this study, we use...")

**Author response: Following scientific convention, we will continue to use past tense in this report of our results.**

Lines 10-12 "We demonstrated that lightweight drones and aerial photogrammetry can be cost effective tools to capture short-term coastal erosion dynamics and related shoreline changes along discrete sections of permafrost coasts." - This goes to the conclusions

**Author response: Correction implemented.**

Figure 1c - remove "Text" from the top right side of the map?

**Author response: Correction implemented.**

Line 17 - please add a reference for Figure 1a

**Author response: Correction implemented.**

Line 20 - "the mean annual air temperature is..."; "the mean annual precipitation is..."

**Author response: Correction implemented.**

Line 22 - "between 2000 and 2011" or "in 2000-2011"

**Author response: Correction implemented.**

Line 24 - delete "in this region"

**Author response: Correction implemented.**

Line 28 – northwesterly and easterly winds; "they exert..." "and with easterly winds facilitating the transport of warm water from the Mackenzie River to Qikiqtaruk Herschel Island" - unfinished phrase? Facilitate?

**Author response: we have rewritten this paragraph for greater clarity.**

Line 1 - sea-level

**Author response: Correction implemented.**

Line 14 - Processing parameters are reported...

**Author response: Correction implemented.**

Line 20 " Río and Gracia, 2013), and survey parameters" -replace by " Río and Gracia, 2013); survey parameters" "and survey parameters and shoreline errors are given in Table 1" - this reference goes to section 3.1 (regarding the survey parameters); the reference to Table 1 in the context of shoreline position errors is repetitive with Lines 15-17.

**Author response: We have revised the manuscript to include reference to Table 1 in section 3.1, and remove unnecessary repetition.**

Line 25 - delete "calculated"

**Author response: Correction implemented.**

Line 5 – Figure 5 should be mentioned after the reference in the text to Figures 2, 3 and 4.

**Author response: Correction implemented.**

Line 12: "by a net total of 143.7 _ 28.4 m" - is it an average value for the whole segment?

**Author response: Yes, sentence reworded for clarity.**

Line 16 "shoreline retreat was 14.5 _ 3.2 m, an average rate of 36 cm per day." - replace by "the shoreline retreated by 14.5 _ 3.2, with an average rate of"

**Author response: Correction implemented.**

5 Line 17 - the shoreline positionS

**Author response: Correction implemented.**

10 Line 18 - meant that THE shorelines

**Author response: Correction implemented.**

15 Lines 19-20 "Coastline retreat was highly episodic in time and space, occurring primarily over two periods" - repetitive, replace by "Coastline retreat primarily occurred over two periods"

**Author response: Correction implemented.**

Line 21: " There was minimal change in coastline position DURING SIX DAYS between August 5th and August 11th

**Author response: Correction implemented.**

Line 25: " a 13-month period"

**Author response: Omitting the conventional hyphen here was deliberate to comply with The Cryosphere's house style ("is our house standard not to hyphenate modifiers containing abbreviated units"). We will continue to follow this house style, but are happy to defer to editorial preference on this.**

Line 26: " in Figure 4, sampled across the A-B-transect indicated on Figure 3." - replace " in Figure 4; they were sampled across the A-B-transect indicated in Figure 3".

**Author response: Correction implemented.**

Page 8 Line 3 - from three to ten days

**Author response: Correction implemented.**

Line 4 -and their speed reached up to...

**Author response: Correction implemented.**

Line 5 "For zero to three days prior to the 1st 5 2017 survey (on 2017-07-06)" replace by " For zero to three days prior to the same survey"

**Author response: Correction implemented.**

Line 11 - of very strong winds

**Author response: Correction implemented.**

Line 16 - and facilitate further undercutting.

**Author response: Correction implemented.**

5  Line 18 – and the wind speed was low.

**Author response: Correction implemented.**

10  Lines 20-21 These meteorological conditions resulted in large waves and undercutting - sounds more logically.

**Author response: Correction implemented.**

Line 29 - where retreat rates typically range.

**Author response: Correction implemented.**

Line 30 - between 0 and 2 m

**Author response: Correction implemented.**

Page 9 Lines 13-14 - "In this case, however, for the total 17.4 m of shoreline retreat between 2016 and 2017 reported here to remain consistent with the long-term average of 2.2 m a -1 , no further erosion of this reach would need to occur for more than seven years" - rephrase: In this case, however, to remain consistent with the long-term average of 2.2 m a -1 , no further erosion of this reach would need to
30  occur for more than seven years after the retreat of 17.4 m in 2016-2017.

**Author response: Correction implemented.**

5 Page 11
Lines 2-3 " Further factors facilitating rapid erosion at this coastal reach ARE the high ice content (ca. 40% Obu et al., 2016) and the low relief"

**Author response: Correction implemented.**

Line 10 -Although this region is microtidal - "although the studied region is microtidal"

**Author response: Correction implemented.**

Page 11,
Lines 13-14 - "Winds exert substantial control over local sea levels, with north-westerly winds driving a positive storm surge and easterly winds driving a negative storm surge (Héquette et al., 1995;
20 Héquette and Barnes, 1990)." - repetitive; already appeared in the Introduction

**Author response: we have rewritten this paragraph to reduce repetition.**

25 Page 12 Line 11 - on Bykovsky Peninsula?

**Author response: Correction implemented.**

30 Figure 2. What is the image at the background? Figures 2, 3 and 4. Would be better readable if you used different colours for coastlines of different time periods instead of shades of grey and black

**Author response: As noted in the figure caption, the background image in Figure 2 is the 2917-07-06 image. We have experimented with a number of approaches the symbology of these shoreline positions, included various combinations of colours and patterns. Unfortunately, the proximity of the lines means that there is substantial overpotting when viewed at most scales. While we found that using colours did not materially improve the legibility of the shoreline positions, we do believe that the greyscale symbology provides sufficient information to readers in this context.**

**Referee #2**

Cunliffe et al. present a case study for an eroding permafrost coastline along the Canadian Beaufort Sea Coast using historic photos, satellite images, and airborne images acquired from a UAV. The imagery ranged in spatial resolution from 3.5 m to 0.02 m and consisted of images acquired between 1952 and 2017 and focused on a 500 m segment of coastline. It appears that the historic imagery was already published previously (could be better clarified in the paper) and the novelty of this paper was the high temporal image acquisition using UAVs in 2017. Seven UAV surveys were used to create high spatial resolution orthophotos and digital surface models to assess coastal change rates on the order of days to weeks during July and August of 2017. The paper is well written and organized. The study design and presentation of results are clear but need to be improved. In particular, the mismatch in image spatial resolution and temporal observations require further consideration in the paper. A number of suggested edits and revisions are provided below to help refine the paper to make it suitable for publication in The Cryosphere.

**Author response: We thank Referee 2 for their positive appraisal of our manuscript and for their constructive suggestions. We have revised our manuscript in light of this feedback (responses to specific points of feedback provided below), and believe that our improvements address the points raised.**

General Comments
The comparison between decadal-scale erosion rates from images with a spatial resolution that ranged from 0.5 m to 3.5 m with coastal change positions determined from images with a spatial resolution of 0.02 to 0.04 m requires further validation . This is particularly important given the assertion that erosion rates in 2017 was 14.5 m/yr compared to a long-term average rate of 2.2 m/yr, or as stated in the abstract more than six times faster. The authors need to include a suitable image from 2017 or 2018 at a resolution that is more in line with image resolutions available historically to demonstrate that the increased resolution of the UAV imagery is not responsible for the measured increase in erosion, simply due to being able to better detect the feature of interest. Doing a quick survey of images available from DigitalGlobe shows that there are some potential options available for the study site in 2017 and 2018 that could provide this necessary check.

**Author response: We agree that error estimation is an important consideration when working with different resolution data. However, we do believe that our comparison between images with different spatial resolutions is appropriate, as these differences in resolution are explicitly described by the 'pixel error' term (Table 1) and propagated through to both the uncertainty in shoreline position (Equation 1), and shoreline retreat rates (Equation 2; Table 2). This treatment of uncertainty in assimilating data sources of different quality is well established (e.g. Irrgang et al., 2018; Radosavljevic et al., 2016; Río and Gracia, 2013). We found that change in shoreline position between the coarse and fine resolution images (between 2011-08-31 – 2016-07-27) was 0.7 ± 0.3 m a-1; so our comparison between these two periods indicates a slower than average rate of change. The very rapid changes in shoreline position we found were in comparisons between fine-grain drone-derived products, and were corroborated by our own field observations. We have added a sentence to our discussion clarifying this: "Our own qualitative observations on the ground over the summer of 2017 (Video S1) confirmed the extremely rapid shoreline changes described above." We do not think that purchasing and analysing additional more coarse-resolution recent imagery would be additionally informative in this instance.**

On line 30 of the abstract the authors report that in 2017 mean coastal retreat was 14.5 m/yr. However, in table 2 it appears that there were only 40 days of erosion analyzed during this period. It appears that the 14.5 m of erosion refers to the magnitude of shoreline change and not an annual rate. This critical point needs to be better clarified and the mismatch in temporal periods among observation periods given more careful consideration . One consideration could be that the image acquired on 2016-07-27 be compared with the image acquired on 2017-08-15 to determine the most recent annual erosion rate instead of using the 2017-07-06 for this. Reporting it in this manner and then using the UAV image acquisitions within this latter annual-scale period to assess event driven erosion patterns and controls might make for a cleaner analysis and presentation of results.

**Author response: Thank you for this feedback, we have revised our manuscript to further clarify that the very rapid erosion we report was measured over a period of just 40 days. In this study, we wanted to test the capabilities of UAV-derived observations to describe intra-seasonal change in shoreline positions, and relate these observations to longer term changes. To achieve this, it is necessary to compare time periods with different lengths, and we believe that we are explicit about this comparison (especially in Table 2). We do report the recent (near) annual rate for 2016-07-27 – 2017-07-06 (21 days less than a year) in Table 2; this omits some of the open water season and is therefore conservative if considered as an annual rate. The period 2016-07-27 – 2017-08-15 would be 19 days more than year, but the additional days are biased to the open water season, thus likely overestimating an annual rate. We have expanded our discussion to state: "Over a 384-day period from 27th July 2016 to 15th August 2017, we observed a large retreat in the shoreline position, with an average of 17.4 m, although note that this period is 19 days longer than a year and includes a disproportionate number of days from the open water season." This 17.4 m value is computed from summing 2.9 m + 14.5 m (the net shoreline change between 2016-07-27 – 2017-07-06, and between 2017-07-06 – 2017-08-15).**

Considering that the historic remote sensing data was apparently previously published (is this what previously analysed refers to on line 23 page 5) the authors need to enhance their methodology and presentation of the imagery acquired with the UAV surveys. The authors should provide information on the altitude of the UAV during image acquisition, the orientation of the flight paths relative to the coastline, why they recommend using front lap and side of 10 and 20 respectively while only using 5

and 10 respectively, the number of ground control points established in the field, and why the authors did not constrain their orthophotos and digital surface models using ground check points when this method is recommended in the literature. All of this should be correctable and is not seen as a major sticking point. The authors are also encouraged to maximize the use of their UAV data by analyzing the digital surface models constructed in Agisoft. Currently this assessment consists of four sentences in the results section. The authors mention that erosion occurring after the fourth UAV survey prevented proper construction of digital surface models in the latter efforts. However, the digital surface model data acquired during the first surveys combined with the shoreline positions digitized from the latter time period orthophoto mosaics should provide sufficient information to add this element to the paper.

**Author response: Thank you for this suggestion and constructive thoughts about how to approach data collection and analyses of landscape change using drones. In our other work, we are strong advocates for the inclusion of ground control markers, sufficient overlap and appropriate methods to facilitate the best-possible 3D model construction (Cunliffe et al. 2016; Cunliffe and Anderson, 2019; Assmann et al. 2018, https://arcticdrones.org/). However, this particular data collection was opportunistic and not a part of our planned data collection. Thus, we were not able to collect data using our prefered method, though we still believe our results are robust given our data collection constraints. We have added additional information describing the UAV surveys, including the range of altitudes and the number of ground control markers uses to constrain each photogrammetric build in Table 1. The orientation of the flight paths varied between surveys, often due to weather (wind) constraints, but we do not believe that flight line orientation relative to the coast make a material difference to this photogrammetric approach. We recommended higher levels of overlap than we used because we wanted to help future users of this approach avoid making the mistake of insufficient overlap, which can have negative implications of image alignment and the geometric stability of photogrammetric reconstructions. Unfortunately, we did not have sufficient ground control markers to allow independent evaluation of the photogrammetric reconstructions. Our qualitative evaluations of the orthomosaic co-registrations indicated that the RMSE errors (Table 1) appeared to be conservative assessments of the registration error. We wanted to highlight best practice in this area, so that future studies would be able to improve upon our data collection. We have also extended our analysis of surface elevation change, measuring removal rates of ca. 0.79 m$^3$ m$^2$ d$^1$ of material (ca. 13,800 m$^3$ over 500 m over 35 days), and have extended the methodology, results and discussion sections of the manuscript accordingly.**

Specific Edits and Questions
- Replace the use of grain with resolution throughout the paper

**Author response: As requested, 'grain' has been replaced with 'resolution'.**

- Consider changing the use of drone to UAV throughout the paper

**Author response: Thank you for this suggestion. In line with the large and growing body of literature on drones in environmental science, we would prefer to continue using the term 'drone' in this manuscript as we feel this term is**

**becoming more dominant in the literature as other terms such as UAV, UAS, RPAS are becoming less frequently used. We believe that our meaning of this term is clear from the text on page two; however, we are happy to defer to Editorial preference regarding this nomenclature in The Cryosphere.**

- Equation 1 seems to be incomplete according to variables presented in Table 1 to determine shoreline change uncertainty. Check this.

**Author response: The input parameters for Equation 1 (now Eq. 2) are present in Table 1, and we have revised the**
10 **wording to increase clarity.**

 - Was the CTD data acquired in 2015 or during the study period in 2017. Check line 7 on page 7. If from 2015 how is it relevant to this study?

**Author response: The CTD data reported was collected during 2017, and this typo has been corrected in the manuscript.**

20 - Specify whether the time-lapse camera in operation for 4 days imaged the study coastline during the observation period.

**Author response: The time-lapse camera was observing the study coastline, and we have amended the manuscript to make this more explicit.**

- Change cm per day on line 16, page 7 to m per day

**Author response: Unit change implemented.**

- Please explain the significance of the linear regression method being more conservative than the end point method as reported on lines 23-25, page 12

**Author response: We have revised this text, which now reads: "Erosion rates from linear regression tend to underestimate rates calculated from end point rates (Dolan et al., 1991; Radosavljevic et al., 2016), which is consistent with our findings of 1.9 m a 1 versus 2.2 m a-1, but linear regression and end point rates alone do not account for uncertainty in shoreline positions (Himmelstoss et al., 2018). Changes in the rate of mean shoreline position for all time points are shown on Figure S4." Figure S4 is a new addition to the Supplementary Information, depicting the differences in average shoreline position over time and the linear regression rate.**

- Adding field photos of the study coast would add useful information to the paper and provide a context for understanding the permafrost characteristics at the site.

**Author response: We agree that photographs (and videos) can be extremely helpful in conveying useful information regarding research subjects. We included such additional information in the Supplementary Information (Figure S3 and Video S1). As these resources would be available with this manuscript, and photographs of this coastline have previously been published in Radosavljevic et al. (2016), we did not think that it would be necessary to include them in the body of the paper. Again, we are very happy to defer to editorial guidance on whether including additional photographs of this site in the manuscript itself would be helpful.**

References cited:
Assmann, J.J., Kerby, J.T., Cunliffe, A.M., Myers-Smith, I.H., 2018. Vegetation monitoring using multispectral sensors - best practices and lessons learned from high latitudes. Journal of Unmanned Vehicle Systems 334730. https://doi.org/10.1101/334730

Carrivick, J. L., Smith, M. W. and Quincey, D. J.: Structure from Motion in the Geosciences, John Wiley & Sons, Ltd, Chichester, UK., 2016.

Cunliffe, A., Anderson, K., 2019. Measuring Above-ground Biomass with Drone Photogrammetry: Data Collection Protocol. Protocol Exchange. https://doi.org/10.1038/protex.2018.134

Cunliffe, A.M., Brazier, R.E., Anderson, K., 2016. Ultra-fine grain landscape-scale quantification of dryland vegetation structure with drone-acquired structure-from-motion photogrammetry. Remote Sensing of Environment 183, 129–143. https://doi.org/10.1016/j.rse.2016.05.019

Irrgang, A. M., Lantuit, H., Manson, G. K., Günther, F., Grosse, G. and Overduin, P. P.: Variability in Rates of Coastal Change Along the Yukon Coast, 1951 to 2015, Journal of Geophysical Research: Earth Surface, 123(4), 779–800, doi:10.1002/2017JF004326, 2018.

James, M.R., Robson, S., Smith, M.W., 2017. 3-D uncertainty-based topographic change detection with structure-from-motion photogrammetry: precision maps for ground control and directly georeferenced surveys. Earth Surf. Process. Landforms 42, 1769–1788. https://doi.org/10.1002/esp.4125

[revised manuscript text omitted]

~~Our observations suggest that the rapid coastal retreat in 2017 may have resulted from multiple factors interacting over several years. On Qikiqtaruk—Herschel Island, atmospheric warming (Burn and Zhang, 2009; Myers-Smith et al., Accepted) has increased the temperature of permafrost (Burn and Zhang, 2009; Myers-Smith et al., Accepted) and deepened the active layer (Myers-Smith et al., Accepted) at locations just ca. 1 km from the study reach. Permafrost temperatures along the study reach may also be influenced by discharge from a creek across part of the alluvial fan (Figure 1); however, long-term discharge records do not exist for this stream. In this area, the onset of seasonal sea-ice melt has moved earlier over the last 18 years, with ice-free seasons lengthening by nine days per decade between 1979-2013 (Stroeve et al., 2014), and summer minimum sea ice concentrations decreasing over the last 39 years (Myers-Smith et al., Accepted). Long-term, the thermo-abrasion of~~

Field Code Changed

permafrost bluffs at this site is likely enhanced by (i) relative sea level rise of 1.1 to 3.5 mm a$^+$ (James et al., 2014; Manson et al., 2005), (ii) earlier ice-break up (Mahoney et al., 2014), (iii) longer open water seasons (Barnhart et al., 2014; Stroeve et al., 2014), (iv) increased wave heights (Barnhart et al., 2014), and (v) increased discharge of warm water from the nearby Mackenzie River . XXXXX

**5.1. The advantages and disadvantages of using drones to quantify short-term coastal erosion dynamics**

The use of drone surveys in this study proved to be an effective tool to measure the dynamics of short-term erosion along this permafrost coastline. Photogrammetric analysis of drone-acquired image data yielded orthomosaics, inferred shoreline positions (Figures 3 and 4), and elevation models (Figure 4) that provide quantitative information on coastal structure. Drone surveys can provide fine spatial grain and accurate measurements of the coastline position at high temporal frequencies, allowing coastline change to be quantified and related to meteorological observations on a supra-annual timescale (Figure 5). Over a 384 day period from July 2016 to August 2017, we were able to observe a very rapid and substantial change in the shoreline position using drone surveys, on average 17.4 m of retreat across the 500 m study reach. Given the episodic nature of coastal retreat, it is difficult to compare short term rate changes with long term observation periods (<2 vs. >10 years, respectively) (Dolan et al., 1991). In this case, however, for the total 17.4 m of shoreline retreat between 2016 and 2017 reported here to remain consistent with the long-term average of 2.2 m a$^+$, no further erosion of this reach would need to occur for more than seven years.

Lightweight drones can be deployed at relatively low cost when suitably trained and equipped personnel are on site. However, the costs of accessing high latitude sites can be substantial, potentially contributing to uneven distributions of monitoring sites (Metcalfe et al., 2018). The temporal resolution of drone surveys can greatly exceed those available by more traditional forms of remote sensing, for example satellite observations or surveys from manned aircraft (Casella et al., 2016; Stow et al., 2004; Whalen et al., 2017). High temporal frequency surveys can provide quantitative insights into erosion processes that vary greatly in time and space, and these quantitative measurements may have stronger physical meaning that previously available proxies, such as the apparent cross sectional area of detached blocks extracted from time-lapse photography (e.g. Barnhart et al., 2014). Surveyable spatial extents are also limited by safety and regulatory restrictions, and depend on the size and the range of the remotely piloted drone. When supplemented by other monitoring of environmental variables (such as wave field and sea surface temperature), such spatial observations could be used to robustly evaluate and subsequently refine process-based numerical models of coastal erosion over multiple temporal scales (Barnhart et al., 2014; Casella et al., 2014; Wobus et al., 2011).

By allowing measurement of the volume and consequently mass of eroded material, digital elevation models can be more informative than simple 2D representations of shoreline position. Digital elevation models were generated following the eight drone surveys (Figure 4). However, faster than anticipated coastal retreat destroyed some ground control markers, resulting in insufficient spatial constraint of the photogrammetric reconstructions of two of the latter surveys (2017-08-05 and 2017-08-15).

5  This weaker constraint contributed to larger elevation errors in these two reconstructions, resulting in the apparent datum shift in Figure 4. If 3D elevation observations are required, care should be taken when deploying ground control markers and conducting drone surveys to ensure that there will be sufficient spatial constraint of the photogrammetric modelling process, even if coastal retreat is faster than expected. For further recommendations on optimising ground control placement, see Carrivick *et al.* (2016) and James *et al.* (2017).

In summary, drone surveys are highly suitable when there is a need to accurately measure small changes (e.g. $\leq$ 0.3 m) in shoreline positions over limited extents (e.g. $\leq$ 5-10 km in length). Fine spatial grain measurements from drone products are especially useful for isolating the drivers of coastal erosion events, and continued miniaturization of thermal and multispectral cameras for drone platforms will create opportunities to better understand these mechanisms of change. While drone surveys

15  can also be used when shoreline position changes are much greater, traditional data sources such as optical satellite observations can be better suited for observing change across larger sections of coastline. High levels of cloud cover in Arctic regions limits the frequency of successful observations (Hope et al., 2004; Stow et al., 2004), but continuing advances in satellite sensors have increased the spatial resolution and revisit frequency of observations. Despite this, freely available products are currently only available for spatial grains of ca. $\geq$ 10 m (e.g. Sentinel 2), and finer-grain (< 4 m) products have

20  non-trivial costs for each scene.

**5.2. Short-term coastal erosion dynamics**

Between the survey in 2016 (2016-07-26) and the first survey in 2017 (2017-07-06), a large portion of beach and cliff debris appeared to have been removed (Figure 4), potentially during storm events in the autumn of 2016 or ice bulldozing during ice-

25  breakup in spring 2017. Removal of this protective material may have increased the susceptibility of these cliffs to rapid erosion in the summer of 2017. The two periods with the most rapid erosion in 2017 (the 27 days between July 13[th] to July 30[th], and the four days between August 11[th] to August 15[th]) were both associated with strong wind events (six-hour moving averages exceeding >40 km h[+]) preceded by relatively high air and water temperatures. Together, these conditions likely enhanced the thermo-abrasional processes undercutting the ice-rich bluff prior to the first survey, creating the conditions for

30  abrupt erosion. Further work relating coastline change to meteorological and oceanographic factors over short timescales would need to further consider the latencies involved between meteorological and oceanographic conditions, undercutting of permafrost cliffs, and planform change as observed from an aerial perspective.

The coastal erosion processes we observed during 40 days of 2017 correspond with the conceptual model described by Barnhart *et al.* (2014) (Video S1 and Figure S3). The bluffs along the alluvial fan were affected by both thermo-denudation but particularly thermo-abrasion due to the combined mechanical and thermal action of sea water causing undercutting and subsequent block failure (Barnhart et al., 2014; Günther et al., 2012). These thermal processes are likely influenced by warm surface waters delivered from the Mackenzie River Delta during easterly wind conditions (Dunton et al., 2006). Further factors facilitating rapid erosion at this coastal reach is the high ice content (ca. 40% Obu et al., 2016) and the low relief, as less material is deposited at the base of the bluff following cliff failure, thus reducing protection of the bluff base from further wave action (Héquette and Barnes, 1990).

Over shorter timescales through the summer of 2017, coastal retreat was highly episodic. The main mode of erosion was block failure driven by thermo-abrasional undercutting, which appeared to be largely influenced by fluctuations in water level combined with wave action. Water level fluctuations appeared to be mainly determined by wind generated surges and waves, superimposed on tidal patterns. Although this region is microtidal, with a mean range of just 0.15 m for semidiurnal and monthly tides, these are superimposed on a ca. 0.66 m annual tidal cycle which peaks in late July (Barnhart et al., 2014), corresponding with our intensive short-term observation period. Annual tides may influence the timing of coastal retreat within the ice-free season in this area. Winds exert substantial control over local sea levels, with north-westerly winds driving a positive storm surge and easterly winds driving a negative storm surge (Héquette et al., 1995; Héquette and Barnes, 1990). The direction and frequency of wind patterns observed in 2017 (Figure 5, Figure S1) are similar to those reported in June-Sept from 2009 to 2012 (Radosavljevic et al., 2016 figure 4 therein). However, overall wind speeds were higher in 2017, with a greater proportion of periods with mean speeds in excess of >30 km h$^{-1}$. During the two periods with highest erosion rates, there were multiple strong storm events with both easterly and north westerly winds with 6-hour average speeds in excess of 30 km h$^{-1}$ and 40 km h$^{-1}$, respectively (Figure 5). We were able to provide quantitative insights into these highly episodic erosion processes, because of the high temporal frequency of shoreline position observations. For example, ca. 30% (4.2 m) of the 14.5 m of shoreline retreat occurring in the summer of 2017 happened in just four days (August 11[th] to August 15[th]), suggesting that discrete storm events can play a major role in permafrost shoreline evolution (Solomon et al., 1993).

**5.3 Long-term pre-conditioning of coastal erosion**

Our observations suggest that the rapid coastal retreat in 2017 may have resulted from multiple factors interacting over several years. On Qikiqtaruk—Herschel Island, atmospheric warming (Burn and Zhang, 2009; Myers-Smith et al., Accepted) has increased the temperature of permafrost (Burn and Zhang, 2009; Myers-Smith et al., Accepted) and deepened the active layer (Myers-Smith et al., Accepted) at locations ca. 1 km from the study reach. Permafrost temperatures along the study reach are likely also be influenced by a creek, which discharges across parts of the alluvial fan (Figure 1); however, long term discharge records do not exist for this stream. In this area, the onset of seasonal sea-ice melt has moved earlier over the last 18 years,

with ice-free seasons lengthening by 9 days per decade between 1979-2013 (Stroeve et al., 2014), and decreasing summer minimum sea ice concentrations over the last 39 years (Myers-Smith et al., Accepted). The combination of relative sea level rise of 1.1 to 3.5 mm a$^{-1}$ (James et al., 2014; Manson et al., 2005), earlier ice-break up (Mahoney et al., 2014) and longer open water seasons (Barnhart et al., 2014; Stroeve et al., 2014), increased wave heights (Barnhart et al., 2014), and increased transport of warm water discharged from the nearby Mackenzie River (Carmack and Macdonald, 2002; Dunton et al., 2006; van Vliet et al., 2013) are long-term factors which likely enhance the thermo-abrasion of permafrost bluffs at this site.

**5.4. Short-term coastal erosion in the context of long-term observations**

The overall rapid coastline retreat observed in this study reach is consistent with, but greater than, earlier analysis of neighbouring coastal reaches on Qikiqtaruk – Herschel Island between 1952 and 2011 (Radosavljevic et al., 2016) and also coastal retreat observed in other Arctic permafrost coastlines (Günther et al., 2013b; Irrgang et al., 2018; Jones et al., 2009a; Whalen et al., 2017). Coastline retreat rates almost doubled from 7.6 m a$^{-1}$ (1955-2009) to 13.8 m a$^{-1}$ (2007-2009) at Cape Halkett on the Alaskan Beaufort Sea (Jones et al., 2009a), and more than doubled from 2.2 m a$^{-1}$ (1952-2010) to 5.3 m a$^{-1}$ (2010-2012) on Bykovsky Island, Siberia (Günther et al., 2013b). Increases in erosion rates greater than two-fold are generally reported for low elevation coasts, such as the one shown in this paper and in (Jones et al., 2009a). On the Yukon Coast, average coastal retreat rates were 0.5 m a$^{-1}$ between 1950-1970 (Harper et al., 1985) and 0.7 m a$^{-1}$ between 1950-2011 (Irrgang et al., 2018), with maximum reported rates of 22 m a$^{-1}$ on Pelly Island (NWT) 130 km to the east along the Yukon NWT Coast (Whalen et al., 2017). Robustly testing whether erosion of permafrost coastlines may be accelerating in this region and more widely will require further analysis of shoreline position change at (near-)annual temporal resolution, considering a larger range of representative coastal reaches and study sites.

Over the 65 years from 1952 to 2017, the coastline in this study reach retreated at a rate of 2.2 m a$^{-1}$, with average rates over decadal periods ranging between 0.7 to 3.0 m a$^{-1}$ (Table 2). This study reach lies within the slightly larger 'coastal reach 3' unit considered by Radosavljevic *et al.* (2016); consequently, differences in reach length and historic image co-registration result in some slight differences between the erosion rates reported herein and those previously reported. Our finding that the overall retreat rate calculated from the linear regression method (1.9 m a$^{-1}$) is more conservative than the rate calculated by the end-point method (2.2 m a$^{-1}$) is consistent with earlier reports (Radosavljevic et al., 2016). In either case, this long-term rate is fast relative to circum-arctic observations, where retreat 
[revised manuscript text omitted]

Mahoney, A. R., Eicken, H., Gaylord, A. G. and Gens, R.: Landfast sea ice extent in the Chukchi and Beaufort Seas: The annual cycle and decadal variability, Cold Regions Science and Technology, 103, 41–56, doi:10.1016/j.coldregions.2014.03.003, 2014.

Mancini, F., Dubbini, M., Gattelli, M., Stecchi, F., Fabbri, S. and Gabbianelli, G.: Using unmanned aerial vehicles (UAV) for high-resolution reconstruction of topography: the structure from motion approach on coastal environments, Remote Sensing, 5(12), 6880–6898, doi:10.3390/rs5126880, 2013.

Manson, G. K., Solomon, S. M., Forbes, D. L., Atkinson, D. E. and Craymer, M.: Spatial variability of factors influencing coastal change in the Western Canadian Arctic, Geo-Mar Lett, 25(2–3), 138–145, doi:10.1007/s00367-004-0195-9, 2005.

Mars, J. C. and Houseknecht, D. W.: Quantitative remote sensing study indicates doubling of coastal erosion rate in past 50 yr along a segment of the Arctic coast of Alaska, Geology, 35(7), 583–586, doi:10.1130/G23672A.1, 2007.

Metcalfe, D. B., Hermans, T. D. G., Ahlstrand, J., Becker, M., Berggren, M., Björk, R. G., Björkman, M. P., Blok, D., Chaudhary, N., Chisholm, C., Classen, A. T., Hasselquist, N. J., Jonsson, M., Kristensen, J. A., Kumordzi, B. B., Lee, H., Mayor, J. R., Prevéy, J., Pantazatou, K., Rousk, J., Sponseller, R. A., Sundqvist, M. K., Tang, J., Uddling, J., Wallin, G., Zhang, W., Ahlström, A., Tenenbaum, D. E. and Abdi, A. M.: Patchy field sampling biases understanding of climate change impacts across the Arctic, Nature Ecology & Evolution, 2(9), 1443, doi:10.1038/s41559-018-0612-5, 2018.

Myers-Smith, I. H. and Lehtonen, S.: NOAA's Arktcic: The Great Flood of July 2016, Team Shrub: Tundra Ecology Lab [online] Available from: https://teamshrub.com/2016/07/22/noaas-arktcic-the-great-flood-of-july-2016/ (Accessed 11 February 2018), 2016.

Myers-Smith, I. H., Grabowski, M., Thomas, H. J. D., Angers-Blondin, S., Daskalova, G., Bjorkman, A. D., Cunliffe, A. M., Assmann, J., Boyle, J., McLeod, E., McLeod, S., Joe, R., Lennie, P., Arey, D., Gordon, R. and Eckert, C.: Eighteen years of ecological monitoring reveals multiple lines of evidence for tundra vegetation change, Ecology Monographs, Accepted.

[revised manuscript text omitted]